# Parameterization Agnostic RL:
## Fine-Tuning Multiple Policy Classes with Actor-Critic RL

## Abstract

Recent advances in learning decision-making policies can largely be attributed to training expressive policy models, largely via imitation learning. While imitation discards non-expert data, offline and/or online fine-tuning via reinforcement learning (RL) can still learn from suboptimal data. However, instantiating RL training of a new policy class often presents a different challenge: most deep RL machinery is co-developed with assumptions on the policy class, resulting in poor performance when the policy class changes. For e.g., SAC utilizes a low-variance reparameterization policy gradient for Gaussian policies, but this is unstable for diffusion policies and intractable for autoregressive (e.g., transformer) categorical policies. To address this issue, we develop an offline RL and online fine-tuning approach called **parameterization-agnostic RL** (*PA-RL*) that can effectively train multiple policy classes, with varying architectures. The basic idea is that a universal supervised learning loss can replace the policy improvement step in RL, as long as it is applied on "optimized" actions. To obtain these optimized actions, we first sample multiple actions from a base policy, and run global optimization (i.e., re-ranking multiple action samples using the Q-function) and local optimization (i.e., running gradient steps on an action sample) to maximize the critic on these candidates. *PA-RL* enables fine-tuning diffusion and autoregressive policies via RL, while improving performance and sample-efficiency compared to existing online RL fine-tuning methods. *PA-RL* allows us to successfully fine-tune diffusion policies and OpenVLA, a 7B parameter generalist robot policy on real robots.

## 1 Introduction

Recent successes in training decision-making policies in a number of domains such as robotics and language agents largely stem from the use of expressive models combined with large-scale imitation-style training (Zitkovich et al., 2023; Chi et al., 2023; Kim et al., 2024), an approach that has been tried and tested in other sub-fields of machine learning, such as vision and NLP (Ouyang et al., 2022). However, we have also realized that training a policy once and freezing it is not good enough for many real-world deployment scenarios, where some adaptation is needed: for example, a robot must adapt its behavior as the surrounding environment or task changes. The hallmark of an adaptation process is in its use of autonomous, non-expert data.

In these cases, imitation alone is not enough to guarantee the most efficient learning and RL provides an appealing alternative. In principle, off-the-shelf RL algorithms could be used to fine-tune any policy. For instance, by running actor-critic RL (Sutton & Barto, 2018), a policy can be trained towards maximizing the Q-function. However, most existing deep RL algorithms entangle the choice of training objectives and algorithm design decisions with the choice of the policy class. For example, soft actor-critic (SAC) (Haarnoja et al., 2018a), the base learner for many offline and online fine-tuning algorithms (Kumar et al., 2020; Nakamoto et al., 2024), has been extensively tuned for Gaussian (and tanh-Gaussian) policies: swapping the policy for a diffusion policy causes instability (Wang et al.). These instabilities can be severe to the extent that much weaker policy extraction techniques, e.g., critic-based re-ranking (Hansen-Estruch et al., 2023) can outperform the complete policy gradient Wang et al., even though theoretically and with other policy classes this is not the case (Fujimoto et al., 2018a; Ghasemipour et al., 2021). Likewise, in order to extend conservative Q-learning (CQL) (Kumar et al., 2020) to autoregressive policies, Chebotar et al. (2023) had to to make many modifications to the loss in the CQL algorithm. Overall, this means that adapting the best policy training methodologies or parameterization from one policy class to another can be chal-

lenging, and depending upon the policy itself, practitioners are forced to choose a weaker algorithm or spend cycles modifying other components of their approach.

In this paper, we tackle this challenge by developing a single offline RL and online fine-tuning approach, which we call parameterization-agnostic RL (**PA-RL**), that works well regardless of the choice of policy class or backbone. We can train any type of policy class and architecture, as long as the policy updates use a supervised learning loss. Now, to perform policy improvement, we propose that the RL algorithm directly optimizes the *action* (instead of policy parameters). Doing so decouples policy improvement from training the parameteric policy, which can now be done via supervised learning, by maximizing the likelihood of "optimized" actions found by policy improvement. To obtain these optimized actions, we first sample from the base policy several times to get multiple action candidates, and then take gradient steps with respect to the value function to improve those actions in the direction of maximizing values. Then these optimized action samples replace the use of samples from the policy in any value-based RL algorithm, and are used to train the policy themselves. Note that while prior work does use supervised losses for policy training, our main contribution is to show that single approach of this sort can effectively train multiple policy classes.

We evaluate **PA-RL** empirically on a number of domains including simulated robotic manipulation tasks and real robots, with Gaussian, diffusion and autoregressive categorical policies based on transformer backbones, on offline RL and offline-to-online RL fine-tuning problems. Our results show that **PA-RL** attains state-of-the-art performance, outperforming the next best fine-tuning approach by 13% in aggregate over various domains. **PA-RL** produces the largest gains on long-horizon tasks that present multimodal offline data distributions (e.g., CALVIN (Mees et al., 2022) in our experiments), where a more expressive policy class beyond standard tanh-Gaussian is necessary for performance. Most notably, **PA-RL** improves diffusion policies on two manipulation tasks by 20-35% within only 1-2 hours of online RL fine-tuning on a *real* WidowX robot. We also show that **PA-RL** is the first RL method to improve 7 billion parameter OpenVLA (Kim et al., 2024) by 75% within 40 minutes of real-world interaction. We also perform a number of ablation experiments.

Our main contribution is **PA-RL**, a *single* approach for offline RL and online fine-tuning policies with different parameterizations and classes via a supervised learning update on optimized actions. The use of a supervised learning loss renders simplicity and universality to our approach. By combining global optimization and local optimization, **PA-RL** is able to effectively train diffusion and transformer policies with offline RL and offline-to-online RL algorithms (Nakamoto et al., 2024; Kostrikov et al.; Ball et al., 2023). To the best of our knowledge, our results are the first to fine-tune diffusion policies (Chi et al., 2023) (both in simulation and in the real-world), and autoregressive categorical transformer policies (in simulation), all via a single actor-critic RL approach.

## 2 RELATED WORK

Contrary to prior belief, recent work (Park et al., 2024) shows that policy learning can be a big bottleneck in RL, especially in offline RL (Levine et al., 2020). One implication is that enhancing the policy extraction step with the most expressive architectures and the best loss functions would be important, but prior works often tailor the RL approach to a specific policy class (e.g., most work has focused on Gaussian policies). In principle, designing effective algorithms for only one policy class can "overfit" resulting in methods that are actually worse for other policy classes. For instance, while algorithms that use Gaussian policies reparameterize the policy gradient (Lillicrap et al., 2015; Haarnoja et al., 2018a; Fujimoto et al., 2018b), doing so for diffusion policies (Wang et al.) or flows (Mazoure et al., 2020) can be quite unstable and requires per-task tuning. Hence, to make a stable algorithm, Hansen-Estruch et al. (2023) resort to Q-function re-ranking on top of a frozen behavior policy, resulting in a somewhat less powerful policy improvement operator (e.g., compared to EMaQ (Ghasemipour et al., 2021), which uses a similar reranking-based policy improvement operator to TD3+BC (Fujimoto & Gu, 2021), which optimizes the policy through the use of full policy gradient and generally performs better). Most offline RL algorithms that use autoregressive categorical transformer policies run conditional (Kumar et al., 2019) or unconditional supervised regression (Janner et al., 2021; Yamagata et al., 2023; Wu et al., 2024), but Park et al. (2024) show that such approaches are unable to extract the best possible policy. In fact, to fine-tune transformer policies directly via offline RL, Chebotar et al. (2023) had to modify value function training.

Motivated by these findings, in this paper, we build a single actor-critic RL algorithm that is effective for fine-tuning arbitrary policy classes, with a focus on diffusion and transformer policies. Related

works that fine-tune diffusion policies include: DPPO (Ren et al., 2024), which uses a two-layer diffusion-specific policy gradient loss, whereas our approach is applicable outside of diffusion policies (Section 5); IDQL (Hansen-Estruch et al., 2023), which only utilizes action re-ranking akin to global optimization in **PA-RL**, but does not distill it into the policy iteratively and hence results in poor fine-tuning performance in our experiments; DIPO (Yang et al., 2023) and DDiffPG (Li et al., 2024), which only utilizes the "action gradient" akin to local optimization in **PA-RL**, but unlike us does so in an online setting, with no pre-training involved; and DQL (Wang et al.), which utilizes the reparameterized policy gradient estimator but is quite unstable in practice, requiring specific checkpoint selection schemes and regularization to succeed, unlike our approach. Psenka et al. learn diffusion policies via score matching, which Ren et al. (2024) find to be quite unstable. Our method outperforms IDQL (Hansen-Estruch et al., 2023), which is one of the most performant methods in this category. We also instantiate our method for fine-tuning autoregressive categorical transformer policies via offline RL and online fine-tuning methods in simulation successfully. To our knowledge, there is no prior work that attempts to fine-tune such models via value-based RL, with the exception of Chebotar et al. (2023), we make no modifications to value function learning.

Methodologically, our method **PA-RL** appears similar to prior approaches that pose "RL as supervised learning", and use weighted or filtered negative log likelihood (NLL) losses for training (Peng et al., 2019; Peters et al., 2010; Peters & Schaal, 2007; Oh et al., 2018; Abdolmaleki et al., 2018). However, note a crucial difference: while these works largely use the dataset or replay buffer action for training via an NLL loss, **PA-RL** samples *new* actions from the policy, optimizes them against the critic, and then trains the policy via NLL on this action. This allows **PA-RL** to make aggressive updates, thus avoiding the "slowness" associated with supervised regression (Tajwar et al.; Kostrikov et al.; Park et al., 2024), while inheriting its simplicity.

Action optimization from **PA-RL** also resembles prior work that uses CEM optimization to obtain actions from a Q-function in the online RL setting (Kalashnikov et al., 2018; Simmons-Edler et al., 2019; Pourchot & Sigaud, 2019), and supervised learning to improve a policy based on the obtained actions (Neumann et al.; Shao et al., 2022). Unlike **PA-RL**, these methods do not make use of offline pre-training to train the proposal distribution, which we show is important in offline RL and online fine-tuning settings since the critic can give erroneous values outside the support of the dataset seen so far (see Figures 12 and 13; initilization from the offline policy is important).

## 3  PROBLEM SETUP AND PRELIMINARIES

We want to find the optimal policy in a Markov Decision Process (MDP) $\mathcal{M} = (\mathcal{S}, \mathcal{A}, P, r, \rho, \gamma)$, where $\mathcal{S}, \mathcal{A}$ are the state and action spaces, $P(s'|s, a)$ and $r(s, a)$ are the dynamics and reward functions, $\rho(s)$ is the initial state distribution, and $\gamma \in (0, 1)$ is the discount factor. Formally, the optimal policy in an MDP, $\pi^* : \mathcal{S} \mapsto \mathcal{A}$ attains the maximal cumulative discounted sum of rewards, denoted by $V^\pi(s) = \mathbb{E}_\pi \left[ \sum_t \gamma^t r(s_t, a_t) | s_0 = s, a_t \sim \pi(s_t), s_{t+1} \sim p(\cdot|s_t, a_t) \right]$. The Q-function of a policy $\pi$ is defined as $Q^\pi(s, a) = \mathbb{E}_\pi \left[ \sum_t \gamma^t r(s_t, a_t) | s_0 = s, a_0 = a, a_{t+1} \sim \pi(s_{t+1}), s_{t+1} \sim p(\cdot|s_t, a_t) \right]$. We use $Q^\pi_\theta$ to denote the estimate of the Q-function of a policy $\pi$ as obtained via a neural network with parameters $\theta$. The action $a$ is a $d$-dimensional continuous vector in $[-1, 1]^d$.

**Problem settings.** We develop our approach for two settings: **(a)** fully offline (Levine et al., 2020) and **(b)** offline-to-online fine-tuning (Nakamoto et al., 2024). In the former setting, we are given access to an offline dataset of experience, $\mathcal{D}_{\text{off}} = \{(s_i, a_i, r_i, s'_i)\}_{i=1}^N$, collected by a behavior policy, $\pi_\beta$, and want to learn a policy that attains best performance using this dataset. In the latter setting, we are supposed to optimize the policy learned offline, say $\pi_{\text{off}}$, using autonomously-collected interaction data in $\mathcal{M}$. More concretely, we aim to obtain the optimal policy with the smallest number of online samples, efficiently. Our approach, **PA-RL** prescribes a single approach to fine-tune policies of different parameterizations / classes (e.g., diffusion, autoregressive transformers).

**Policy parameterizations.** In our experiments, we consider fine-tuning two types of policy classes: diffusion and transformer policies. Diffusion policies use a conditional Denoising Diffusion Probabilistic Model (DDPM, Ho et al. (2020)) to represent the distribution over action conditioned on the state. A DDPM trains a diffusion step-dependant ($t$) denoising model, $\varepsilon_\phi(a, t|s)$ that is trained with:

$$\mathcal{L}^{\text{ddpm}}(\phi) = \mathbb{E}_{t \sim \mathcal{U}(1, K), \epsilon \sim \mathcal{N}(0, I), (s, a) \sim \mathcal{D}} \left[ \| \epsilon - \epsilon_\phi(\sqrt{\bar{\alpha}_i} a + \sqrt{1 - \bar{\alpha}_i} \epsilon, s, t) \| \right] \quad (3.1)$$

where, given a fixed variance schedule $\beta_1, \ldots, \beta_K$ for the forward diffusion process, $\alpha_t$ is defined as $1 - \beta_t$, and $\bar{\alpha}_t$ as $\prod_{s=1}^K \alpha_s$. To obtain the final action, we start with a random sample $a_K \sim \mathcal{N}(0, I)$,

and iteratively denoise the sample such that $a_{t-1} = \frac{1}{\sqrt{\alpha_t}}\left(a_t - \frac{1-\alpha_t}{\sqrt{1-\bar{\alpha}_t}}\varepsilon_\phi(a_t, s, t)\right) + \sqrt{\beta_t}z$, where $z \sim \mathcal{N}(0, I)$ if $t > 1$ and 0 otherwise, for $K$ total denoising steps. We also fine-tune transformer-based policies that represent the policy $\pi_\phi(a|s)$ as a product of conditional categorical distributions:

$$\pi_\phi(a|s) = \Pi_{i=1}^{d-1}\pi_\phi(\text{tokenize}(a_i)|s, a_{0:i-1}).\tag{3.2}$$

**Offline RL and online fine-tuning methods.** The approach we build only affects policy optimization, and retains the same training procedure for the critic as the base algorithm. Our experiments will focus on two classes of actor-critic based online fine-tuning algorithms (Park et al., 2024): **(1)** algorithms that decouple critic updates from actor updates (e.g., Implicit Q-Learning, IQL (Kostrikov et al.)), and **(2)** algorithms that sample from the actor to train the critic (e.g., Calibrated Q-Learning, Cal-QL (Nakamoto et al., 2024)). Briefly, Cal-QL trains the Q-function to reduce temporal-difference (TD) error, with an additional regularizer that penalizes the learned Q-values on out-of-distribution (OOD) actions as long as Q-values are higher than $V^\mu(s)$, the values of a reference policy, while compensating for this pessimism on actions seen within the training dataset. The Cal-QL critic training objective is given by:

$$\mathcal{L}_Q^{\text{Cal-QL}}(\theta; \phi) = \alpha\left(\mathbb{E}_{s\sim\mathcal{D}, a\sim\pi_\phi(\cdot|s)}\left[\max(Q_\theta(s, a), V^\mu(s))\right] - \mathbb{E}_{s, a\sim\mathcal{D}}\left[Q_\theta(s, a)\right]\right)\tag{3.3}$$

$$+\frac{1}{2}\mathbb{E}_{s, a, s'\sim\mathcal{D}}\left[(Q_\theta(s, a) - \mathcal{B}^\pi\bar{Q}(s, a))^2\right].$$

Where $Q_\theta$ is the learned critic, $\bar{Q}$ is the delayed target Q-function, and $\mathcal{B}^\pi\bar{Q}(s, a)$ is the backup operator: $\mathcal{B}^\pi Q(s, a) = r(s, a) + \gamma\mathbb{E}_{a'\sim\pi(a'|s')}[\bar{Q}(s', a')]$. Computing this loss requires sampling actions from the learned policy $\pi_\phi(\cdot|s)$, which is now an expressive policy class. In contrast, IQL trains the Q-function to regress to a higher expectile of the value function, without needing to query any new action samples from the learned policy (where $V_\psi(s)$ is the value network).

$$\mathcal{L}_V^{\text{IQL}}(\psi) = \mathbb{E}_{(s, a)\sim\mathcal{D}}\left[L_2^\tau(Q_{\hat{\theta}}(s, a) - V_\psi(s))\right]\tag{3.4}$$

$$\mathcal{L}_Q^{\text{IQL}}(\theta) = \mathbb{E}_{(s, a, s')\sim\mathcal{D}}\left[(r(s, a) + \gamma V_\psi(s') - Q_\theta(s, a))^2\right]\tag{3.5}$$

Where $L_2^\tau(u) = |\tau - \mathbb{1}(u < 0)|u^2$ is the expectile loss, and $\hat{\theta}$ are the target parameters for the Q-function. Prior algorithms that fine-tune diffusion policies largely do not apply to transformer policies as they make design choices specific to the diffusion process: for example, Ren et al. (2024) exploits the structure of diffusion; Wang et al. cross-validates against the DDPM loss.

## 4 *PA-RL*: Training Multiple Policy Classes with Actor-Critic RL

Our approach aims to fine-tune multiple policy classes with RL, regardless of scale or parameterization, stably and efficiently. An approach to attain sample-efficient policy improvement is to use an off-policy RL algorithm, which typically alters between fitting a Q-function and updating the policy parameters in the direction of larger predicted Q-values. Typically, value learning treats the policy as a black-box that provides actions for computing and opti-

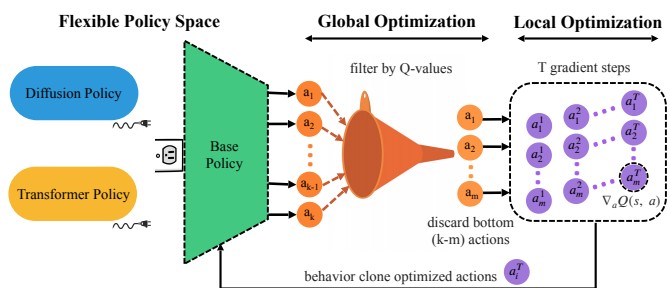

Figure 1: *An overview of* **PA-RL.** Instead of directly passing critic gradients through the policy parameters, *PA-RL* first "optimizes" actions via critic re-ranking and gradient ascent. Then, it trains the policy to mimic the most optimized action.

mizing the Bellman update. Policy improvement, on the other hand, requires optimizing the value function with respect to the policy parameters. Most continuous-action actor-critic RL algorithms estimate the gradient $\nabla_\phi Q(s, \pi_\phi(s))$ with respect to the parameters of the policy $\phi$ for this purpose. Unfortunately, estimating this gradient is quite challenging for most policy classes. For large diffusion policies propagating the policy gradient through the denoising chain can be unstable, often requiring extensive per-environment tuning of hyperparameters (Wang et al.) or truncating the gradient propagation after a subset of denoising steps (Ren et al., 2024). Similarly, auto-regressive policies operate on discrete action tokens, so we must utilize a high-variance REINFORCE (Williams, 1992) policy gradient to optimize the policy.

**Can we devise a simple yet universal approach to policy optimization in offline RL and online fine-tuning?** In order for an approach to be universal across parameterizations, one natural point is to modify policy training to use a negative log likelihood (NLL) loss from supervised learning, since most deep learning machinery is built around optimizing this loss (or a functional approximation). To be able to do so, our method (Fig. 1) builds on the insight that policy improvement can be performed via a supervised learning loss (Neumann et al.; Shao et al., 2022), as long as the loss is applied on *optimized* actions. Thus, we can decompose the policy improvement step into two stages: **(1)** directly optimizing action samples produced by the policy, and **(2)** training the policy to imitate these "optimized" actions. This decomposition avoids having to compute $\nabla_\phi Q(s, \pi_\phi(s))$, or estimate high-variance policy gradient estimates. We would expect this approach to inherit appealing scaling, reliability, and tuning properties of supervised learning losses. In this section, we will detail each of the two stages of the decomposition, and then describe the resulting algorithm.

### 4.1 ACTION OPTIMIZATION

Given a state $s$, a policy $\pi_\phi(\cdot|s)$ checkpoint that appears in the process of learning, and a fixed Q-function $Q_\theta(s, a)$, the objective of this stage is to obtain an action sample that optimizes the Q-function as much as possible, while staying close to the support of seen actions at state $s$. We use $\pi_\phi(\cdot|s)$ as an initializer for the action optimization procedure. In the offline setting, doing so allows us to find the best action close to the support at the current state (and wihtin the support of actions at the current state for a pessimistic algorithm). During fine-tuning, this enables us to still leverage priors learned by the offline policy while adapting it to maximize returns on the task.

To produce an optimized action, we utilize a combination of different types of *action optimization* procedures. First, we consider **global** optimization or sampling that samples multiple actions from the pre-trained policy and discards all but top few actions with highest Q-values under the critic (for computational efficiency). Let $\mathcal{A}_{\pi_\phi,k}(s) := \{a_0, a_1, \cdots, a_{k-1}\} \sim \pi_\phi(\cdot|s)$ denote $k$ sampled actions from the policy. And let $\widetilde{\mathcal{A}}_{\pi_\phi,k}(s) := \{a[0], a[1], \cdots, a[k-1]\}$ denote the set $\mathcal{A}_{\pi_\phi,k}(s)$ with actions put in order of their ranking obtained from the Q-function, i.e., $Q_\theta(s, a[i]) \geq Q_\theta(s, a[j])$, for $i \leq j$. Then, global optimization retains the following subset:

$$\widetilde{\mathcal{A}}_{\pi_\phi,m}(s) = \{a[0], a[1], \cdots, a[m-1]\}, \quad m \leq k. \quad \textbf{(global optimization)} \quad (4.1)$$

Given this subset of the top $m$ actions at a state $s$, we now **locally** improve each action "particle", by performing gradient steps on the action in the direction of the gradient of the Q-function, directly, without changing the policy parameters at all. This sort of a fine-grained local optimization is complementary to the fairly coarse global optimization procedure above as it perturbs the action to another one in its vicinity. Formally, given an action sample $a[i]$, we run $T$ steps of gradient ascent starting from $a^0[i] := a[i]$ to obtain the locally optimal action, $a^T[i]$ as shown below.

$$\text{for } j = 0, \cdots, T-1, \ \ a^{j+1}[i] = a^j[i] + \alpha \nabla_a Q_\theta(s, a)\big|_{a=a^j[i]}, \quad \textbf{(local optimization)}, \quad (4.2)$$

where $\alpha$ is an appropriate learning rate that we choose for optimization. Applying both of these steps enables action optimization to leverage complementary benefits of both of these steps, while avoiding failure modes of either approach (e.g., being trapped in local minima vs not being fine-grained enough). Concretely, let us denote the action set obtained by running local optimization on $\widetilde{\mathcal{A}}_{\pi_\phi,m}(s)$ as $\widetilde{\mathcal{A}}^T_{\pi_\phi,m}(s)$. A pseudocode for action optimization is in Algorithm 1.

### 4.2 POLICY TRAINING VIA SUPERVISED LEARNING

The second stage of **PA-RL** distills optimized actions into the learned policy model. Crucially, this distillation is performed via standard likelihood maximization procedures from supervised learning that most deep learning models are trained to do. While the most direct option is to simply take the action from the set $\widetilde{\mathcal{A}}^T_{\pi_\phi,m}(s)$ that attains the highest Q-value (say, $a^*(\pi, m, T, s)$) and maximize its likelihood under the learned policy $\pi_\phi(\cdot|s)$, another alternative is to distill all action samples from $\widetilde{\mathcal{A}}^T_{\pi_\phi,m}(s)$, but weight the contributions of different actions using the Q-value. We prescribe a simple strategy to choose between these methods (Appendix B.1). To accomplish this, we define a categorical policy distribution over the optimized action samples:

$$\pi_\phi^{\text{Opt}}(a|s, m) := \mathbb{I}\left[a \in \widetilde{\mathcal{A}}^T_{\pi_\phi,m}(s)\right] \cdot \frac{\exp(Q_\theta(s, a))}{\sum_{a' \in \widetilde{\mathcal{A}}^T_{\pi_\phi,m}(s)} \exp(Q_\theta(s, a'))}, \quad (4.3)$$

and train the policy $\pi_\phi(\cdot|\cdot)$ to match this distribution. To do so, we annotate all states in the dataset (including the replay buffer in online fine-tuning) with an action sample from $\pi_\phi^{\mathrm{Opt}}(a|s, m)$, and maximize the likelihood of these actions under the policy, following best practices for supervised learning on this policy class. Formally, we denote this dataset of optimized actions as:

$$\mathcal{D}_{(\phi,\theta,m)}^{\mathrm{Opt}} = \{(s_i, \tilde{a}_i^{\mathrm{Opt}}\}, \ \tilde{a}_i^{\mathrm{Opt}} \sim \pi_\phi^{\mathrm{Opt}}(a|s_i, m). \tag{4.4}$$

For instance, if the policy $\pi_\phi$ is parameterized as a diffusion model, we follow the DDPM (Ho et al., 2020) behavior cloning (BC) objective, and train the policy to predict noise:

$$\mathcal{L}_{\mathrm{policy}}^{\mathrm{ddpm}}(\phi; \theta) = \mathbb{E}_{t \sim \mathcal{U}(1,T), \epsilon \sim \mathcal{N}(0,I), (s,a) \sim \mathcal{D}_{(\phi,\theta,m)}^{\mathrm{Opt}}} \left[ \|\epsilon - \epsilon_\phi(\sqrt{\bar{\alpha}_i}a + \sqrt{1 - \bar{\alpha}_i}\epsilon, s, t)\| \right] \tag{4.5}$$

By using this loss instead of the reparameterized Q-function gradient, we avoid ever backpropagating through the denoising chain, and instead supervise every step of the chain independently. For auto-regressive transformer policies, we use cross-entropy loss objective for next-token prediction.

Finally, we would like to note that while prior work does explore supervised learning losses for training policies (Peng et al., 2019; Abdolmaleki et al., 2018; Oh et al., 2018), the crucial differences between *PA-RL* and these prior techniques stem from the fact that: **(a)** action samples are drawn from the *current* policy, instead of a previous policy or a behavioral policy (Peng et al., 2019), **(b)** local optimization and global optimization employed by *PA-RL* enable aggressive updates on action samples to draw them to novel regions that are otherwise not possible with non-parametric methods that operate on the space of actions directly. While these differences might appear small, we show in our experiments that they have a substantial impact on downstream efficiency of RL training.

### 4.3 PUTTING IT ALL TOGETHER: FINAL *PA-RL* ALGORITHM

*PA-RL* can be used to replace the policy improvement step in multiple RL algorithms. In our experiments, we primarily focus on online fine-tuning and adaptation of offline RL. Hence, we instantiate *PA-RL* using two popular RL fine-tuning methods: Cal-QL (Nakamoto et al., 2024) and IQL (Kostrikov et al.). *PA-RL* only modifies the policy improvement step of each of these methods, while keeping the critic training as it is. Since IQL training does not utilize policy backups, using *PA-RL* in conjunction with IQL is straightforward: simply replace the advantage-weighted regression (AWR) update with the above supervised learning update (e.g., Equation 4.5 for diffusion policies). On the other hand, for Cal-QL and other actor-critic algorithms, where the policy $\pi_\phi(\cdot|s)$ is used to generate action samples for performing the TD-backup, we utilize the optimized action set $\widetilde{\mathcal{A}}_{\pi_\phi,m}^T$ for the Bellman backup. Formally, this means that instead of computing Bellman targets using an updated $\pi_\phi$, we simply compute targets using the optimized policy $\pi_\phi^{\mathrm{Opt}}(\cdot|\cdot, m)$ (Equation 4.3) for Cal-QL. A pseudocode of the algorithm along with the corresponding changes in red is shown in Algorithm 2.

**Implementation details.** We provide a detailed list of hyperparamters and best practices for running *PA-RL* in Appendix B.1. In our experiments, we run *PA-RL* with both state-based and image-based environments, where we utilize best design practices for the critic (Kumar et al., 2022). We also find that additionally including the action $a$ appearing at a given state in the dataset into action optimization can sometimes be helpful. Finally, since native gradient ascent for local optimization is not guaranteed to improve the Q-value for a larger than ideal step size, we only execute a local update if it increases the Q-value after that step.

**Conceptual comparison of *PA-RL* with filtered BC or advantage-weighted regression (AWR).** We now list down a condition under which *PA-RL* optimizes the Q-function better than AWR. Concretely, we show that using a combination of local and global optimization, *PA-RL* is able to improve the policy to a larger extent than AWR. Formally, consider a single state and $\widetilde{\mathcal{A}}_{\pi_\phi,m}(s)$ from Equation 4.1. If local optimization is run for $T$ steps, with a step size $\alpha$, then the Q-values of actions under the optimized policy $\pi_\phi^{\mathrm{Opt}}(a|s, m)$ is given as the left hand side of Equation 4.6. With no local optimization at all (or when the Q-function is used to filter actions in the data as in AWR), the resulting Q-value of the optimized action is given by the right hand side of Equation 4.6. It is easy to see that with high probability, when either $T$ or $m$ or both are large, this inequality holds. Thus, we expect *PA-RL* to generally lead to aggressive updates over AWR.

$$\text{with high prob,} \quad \max_{i=1,2,\cdots,m} \left( Q(s, a_i) + \alpha T \mathbb{E}_t \left[ \|\nabla_a Q(s, a_i^t)\|_2^2 \right] \right) \geq \max_{i=1,2,\cdots,m} Q(s, a_i). \tag{4.6}$$

In practice though, AWR simply upweights the dataset action so it should be less aggressive than RHS of Equation 4.6. Obtaining the LHS of Equation 4.6 requires Taylor's expansion at every step of local optimization, under the assumption that step size $\alpha$ is small enough.

---

**Algorithm 1** Action Optimization $\pi_{(\phi,\theta)}^{\mathrm{opt}}$

---

**Require:** base policy $\pi_\phi$, Q-function $Q_\theta$
 1: Sample actions from $\pi$ to obtain $\mathcal{A}_{\pi_\phi,k}(s)$.
 2: Run global optimization for every state $s$ to retain top $m$ actions, $\widetilde{\mathcal{A}}_{\pi_\phi,m}(s)$
 3: **for** $a$ in $\widetilde{\mathcal{A}}_{\pi_\phi,m}(s) \cup \{a_{\mathrm{data}}(s)\}$ **do**
 4:  **for** i in $\{1, \ldots, \mathrm{T}\}$ **do**
 5:   $a^{(i)} \leftarrow a^{(i-1)} + \alpha \nabla_a Q_\theta(s, a^{(i-1)})$
 6:   **if** $Q_\theta(s, a^{(i)}) \leq Q_\theta(s, a^{(i-1)})$ **then**
 7:    $a^{(i)} \leftarrow a^{(i-1)}$
 8:   **else**
       Break
 9: **return** $\pi_{(\phi,\theta)}^{\mathrm{opt}}$ computed via Equation 4.3

---

**Algorithm 2** Cal-QL + *PA-RL*

---

**Require:** BC loss $\mathcal{L}_{\mathrm{policy}}$, e.g. $\mathcal{L}_{\mathrm{policy}}^{\mathrm{ddpm}}$
 1: Pre-train policy $\pi_\phi$ via offline RL / BC
 2: Initialize Q-function $Q_\theta$
 3: **for** step $t$ in $\{1, \ldots, \mathrm{M}\}$ **do**
 4:  Train Q-function using Eq. 3.3, but use optimized actions for TD targets

$$\theta_t = \theta_{t-1} - \eta_Q \nabla_\theta \mathcal{L}_Q^{\mathrm{Cal\text{-}QL}}(\theta; \phi)$$

 5:  Distill optimized actions to policy

$$\phi_t = \phi_{t-1} + \eta_\pi \nabla_\phi \mathcal{L}_{\mathrm{policy}}(\phi; \theta)$$

 6:  **Collect new online rollouts:**
 7:   $a_t \sim \pi_{(\phi,\theta)}^{\mathrm{opt}}; s_{t+1} \sim p(s_{t+1}|s_t, a_t)$
 8:   $\mathcal{D} \leftarrow \mathcal{D} \cup \{(s_t, a_t, r(s_t, a_t), s_{t+1})\}$

---

## 5 EXPERIMENTAL EVALUATION

The goal of our experiments is to understand the efficacy of *PA-RL* in fine-tuning policies of various parameterizations and classes via RL. To this end, we evaluate *PA-RL* and several prior approaches, on a number of benchmark domains that require learning policies from static offline data (offline RL (Levine et al., 2020)) and then fine-tuning them with limited online interaction in the MDP (offline-to-online fine-tuning (Nair et al., 2020)). Then, we will also present results validating the efficacy of *PA-RL* on two real-robot manipulation tasks and show OpenVLA fine-tuning results with PA-RL in Appendix C. Finally, we perform ablation experiments to understand the utility of different components of *PA-RL*. We first describe our main results and then present ablations.

### 5.1 RESULTS: SIMULATED BENCHMARKS FROM STATE AND IMAGE OBSERVATIONS

We first compare *PA-RL* with prior methods on several benchmark tasks from the D4RL (Fu et al., 2020) suite. Since we report performance in both the offline RL and offline-to-online RL settings, we apply *PA-RL* on top of Cal-QL (Nakamoto et al., 2024) and IQL (Kostrikov et al.), two common offline RL and offline-to-online fine-tuning algorithms, although majority of our results use Cal-QL. We first demonstrate the efficacy of *PA-RL* in training diffusion policies, and compare it to methods that also train diffusion policies. Specifically, we compare *PA-RL* to: **(1)** Implicit Diffusion Q-Learning (IDQL, Hansen-Estruch et al. (2023)), which extends IQL to use diffusion policies via critic-based reranking; **(2)** Diffusion Policy Policy Optimization (DPPO, Ren et al. (2024)), which fine-tunes diffusion policies learned via imitation learning using PPO; and **(3)** Diffusion Q-Learning (DQL, Wang et al.), which trains diffusion policies via a reparameterized policy gradient estimator akin to standard SAC (Haarnoja et al., 2018b).

We study: **(1)** AntMaze tasks from D4RL (Fu et al., 2020) that require controlling the joints of a quadruped ant to reach a goal location in four different maze layouts with a sparse reward; **(2)** FrankaKitchen tasks from D4RL (Gupta et al., 2020), which require solving a sequence of four manipulation tasks in a kitchen environment with a 9-Dof Franka robot; and **(3)** CALVIN benchmark (Mees et al., 2022; Shi et al., 2023) (D → D, with distractor objects), which requires solving a sequence of four manipulation tasks in a tabletop environment directly from visual observations and with human-teleoperated play data. This offline data presents fairly low action coverage but pretty high coverage over different modes of semantic behavior. Due to the diversity of offline data, we believe the CALVIN should stress test the ability of any approach in effectively utilizing the multi-modal nature of diffusion policies for improving efficiency of fine-tuning. All of these tasks present long horizons; and the FrankaKitchen and CALVIN tasks require chaining skills.

***Results:* PA-RL *significantly improves learning efficiency and asymptotic performance of Cal-QL with diffusion policies.*** We compare different approaches for offline RL training and online fine-

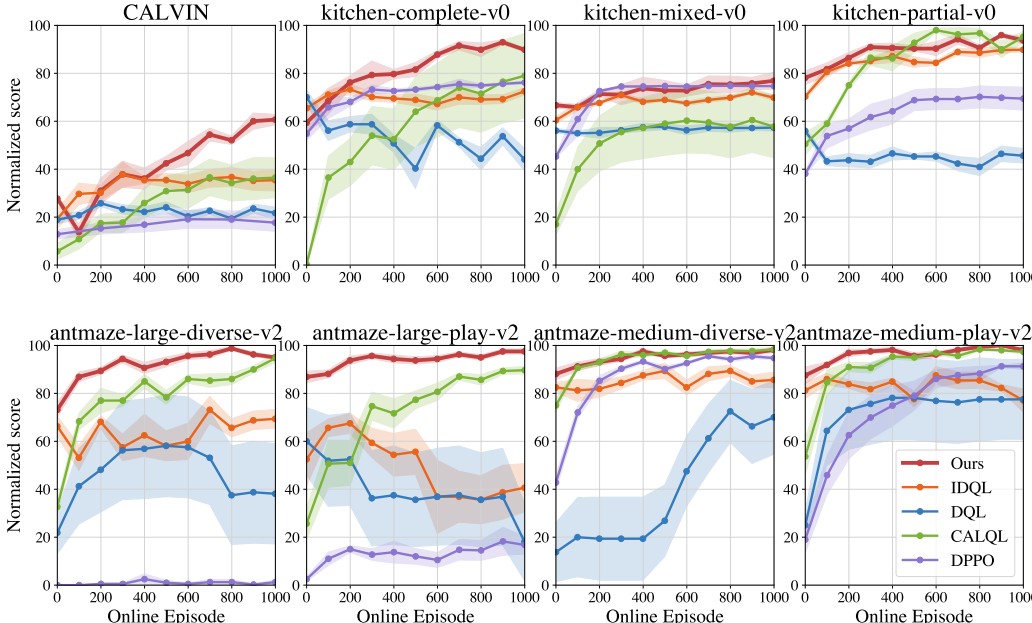

Figure 2: **Learning curves of online fine-tuning** with various methods. Observe that *PA-RL* + Cal-QL (red) largely always dominates or attains similar performance to the next best method. Other methods for fine-tuning diffusion policies (IDQL, DQL, DPPO) are a bit unstable, and perform substantially worse. Since DPPO is substantially more data inefficient, we plot it with different x-axis units: for kitchen each unit is 500 episodes (axis goes from 0 to 500k), for antmaze each unit is 100 episodes (axis goes from 0 to 100k) and for calvin each unit is 10 episodes (axis goes until 10k).

| Domain / Task | IDQL | DQL | DPPO | Cal-QL | *PA-RL* + Cal-QL (Ours) |
|---|---|---|---|---|---|
| *CALVIN* | $19 \rightarrow 35$ | $19 \rightarrow 22$ | $13 \rightarrow 18$ | $6 \rightarrow 36$ | $28 \rightarrow 61$ |
| *Kitchen* (-v0) | | | | | |
| complete | $65 \rightarrow 72$ | $70 \rightarrow 44$ | $55 \rightarrow 76$ | $19 \rightarrow 57$ | $59 \rightarrow 90$ |
| mixed | $60 \rightarrow 70$ | $56 \rightarrow 57$ | $45 \rightarrow 75$ | $37 \rightarrow 72$ | $67 \rightarrow 77$ |
| partial | $70 \rightarrow 90$ | $56 \rightarrow 46$ | $38 \rightarrow 69$ | $59 \rightarrow 84$ | $78 \rightarrow 94$ |
| *Antmaze* (-v2) | | | | | |
| large-diverse | $66 \rightarrow 69$ | $22 \rightarrow 38$ | $0 \rightarrow 1$ | $33 \rightarrow 95$ | $73 \rightarrow 95$ |
| large-play | $53 \rightarrow 41$ | $60 \rightarrow 18$ | $2 \rightarrow 17$ | $26 \rightarrow 90$ | $87 \rightarrow 98$ |
| medium-diverse | $83 \rightarrow 86$ | $14 \rightarrow 70$ | $43 \rightarrow 95$ | $75 \rightarrow 98$ | $88 \rightarrow 98$ |
| medium-play | $81 \rightarrow 77$ | $25 \rightarrow 78$ | $19 \rightarrow 91$ | $54 \rightarrow 97$ | $88 \rightarrow 98$ |
| **Aggregate** | $497 \rightarrow 540$ | $322 \rightarrow 373$ | $215 \rightarrow 442$ | $309 \rightarrow 629$ | $568 \rightarrow 711$ |

Table 1: **Offline-to-online fine-tuning on simulated benchmarks**. *PA-RL* + Cal-QL outperforms every other approach in aggregate, both in terms of the offline performance (left of $\rightarrow$) and performance after 1k episodes of fine-tuning (right of $\rightarrow$). This indicates the efficacy of *PA-RL* in fine-tuning diffusion policies effectively.

tuning in Table 1 and present corresponding learning curves in Figure 2. First, observe that *PA-RL* attains higher offline performance than other methods that use diffusion policies, as well as standard Cal-QL with a tanh-Gaussian policy. Fine-tuning from the offline RL policy learned by *PA-RL* also leads to the best fine-tuned performance in aggregate across all the methods. Concretely, the fine-tuning performance of *PA-RL* is 13% higher than the next best method. On the hardest CALVIN task (where we must learn to control policies from raw visual observations), *PA-RL* attains a **69% improvement** over the *next best* method. This perhaps hints at the efficacy of *PA-RL* in effectively leveraging the increased capacity and expressive power of diffusion policies. Diving deeper, the learning curves in Figure 2 reveal a much stronger trend: the performance of *PA-RL* largely stays above the performance of all other methods throughout training. This indicates the efficacy of *PA-RL* in effectively utilizing the expressivity of diffusion policies during fine-tuning. We also evaluate *PA-RL* in conjunction with IQL on the FrankaKitchen tasks in Table 2, and observe that *PA-RL* + IQL also outperforms standard IQL. This indicates that *PA-RL* is broadly effective.

| Task | tanh-Gaussian RLPD @ 200 | Diffusion *PA-RL* + RLPD @ 200 | Gaussian IQL @ 1k | Diffusion . *PA-RL* + IQL @ 1k | tanh-Gaussian Cal-QL @ 1k | Transformer *PA-RL* + Cal-QL @ 1k |
|---|---|---|---|---|---|---|
| partial | $0 \rightarrow 18$ | $58 \rightarrow 73$ | $40 \rightarrow 60$ | $62 \rightarrow 75$ | $59 \rightarrow 84$ | $33 \rightarrow 95$ |
| mixed | $0 \rightarrow 14$ | $58 \rightarrow 58$ | $48 \rightarrow 48$ | $69 \rightarrow 73$ | $37 \rightarrow 72$ | $42 \rightarrow 84$ |
| complete | $0 \rightarrow 34$ | $70 \rightarrow 81$ | $57 \rightarrow 50$ | $63 \rightarrow 88$ | $19 \rightarrow 57$ | $8 \rightarrow 90$ |

Table 2: **Combining *PA-RL* with different policy parameterizations and critic learning algorithms.** In the hybrid RL setting, ***PA-RL* + RLPD** is able to effectively improve a pre-trained diffusion policy without requiring pre-training the critic. ***PA-RL* + IQL** attains a similar performance on the FrankaKitchen domain as IDQL, proving our method can work with different objectives for the critic. **Transformer *PA-RL*** improves an auto-regressive transformer 224%. To the best of our knowledge, this is the first time an auto-regressive transformer was improved with the Actor-Critic architecture.

***Results:* PA-RL *with hybrid RL.*** Next, we run *PA-RL* on top of RL with Prior Data (RLPD Ball et al. (2023)), a method that incorporates offline data into an online RL training run but does not use offline RL pre-training. In this case, we replace the standard tanh-Gaussian policy in RLPD with a diffusion policy and keep the critic randomly initialized. As shown in Table 2 (left), observe that *PA-RL* is able to improve upon the imitation-learning performance of the diffusion policy after 200 episodes to substantially better performance values than when a Gaussian policy is used for training itself. This further corroborates the efficacy of *PA-RL* in leveraging expressivity of the policy architecture to do sample-efficient learning in the setting of online RL with offline data.

**Results: *PA-RL* + Cal-QL with autoregressive categorical policies.** Our experiments so far evaluate the efficacy of *PA-RL* in fine-tuning diffusion policies. Our next results show that *PA-RL* is also effective in training transformer-based policies that model the distribution over actions autoregressively using categorical distributions. Concretely, this type of policy discretizes each dimension of the action space independently into a set of 128 bins, and then trains an autoregressive model over this sequence of discrete per-dimension action tokens. Observe in Table 2 (right) that *PA-RL* is also able to effectively improve autoregressive categorical policies with Cal-QL, and attains performance 26% better than using tanh-Gaussian policies on average across the three tasks considered. This establishes the efficacy of *PA-RL* in fine-tuning policies of multiple classes.

## 5.2 Results: RL Fine-Tuning of Robot Policies in the Real World

We now show that *PA-RL*, *can* enable fine-tuning diffusion policies on a real robot, resulting in substantial improvements in success rates of the pre-trained policy initialization within just 30 minutes to 2 hours (i.e., 30-70 episodes) of real-world autonomous interaction. To our knowledge, *this is one of the first results to effectively fine-tune diffusion policies on a real robot with actor-critic RL.*

**Real-world robot and task setup.** We study two manipulation tasks (Figures 3 and 7) on a WidowX-250 robotic arm with six degrees of freedom and a single third-person mounted camera. Our setup is inspired by Ebert et al. (2022); Walke et al. (2023) and the policy controls the end-effector pose at a frequency of 5 Hz. The tasks are: **(a)** *"cup to drying rack"*, which requires grasping a plastic cup and placing it in the drying rack across the sink; and

| Task | DDPM (offline) | Iterated Filtered BC | Cal-QL + *PA-RL* (offline $\rightarrow$ online) |
|---|---|---|---|
| Cup to Rack | 50% | 50% | $55\% \rightarrow 90\%$ |
| Pot to Sink (w/ dist. shift) | 50% | - | $80\% \rightarrow 100\%$ |

Table 3: **Real-robot fine-tuning results for *PA-RL*. *PA-RL*** improves the performance of an offline pre-trained diffusion policy on two real robot tasks. Notably, while iterating filtered BC, a simple and stable approach for fine-tuning, does not meaningfully improve over fine-tuning on task **(a)**, *PA-RL* improves substantially. *PA-RL* is similarly effective on task **(b)** under distribution shift.

**(b)** *"pot to sink"*, which requires picking and moving a toy pot from the drying rack to the sink. For task **(a)** the sink contains distractor objects and for both tasks, the positions and rotation of the target object are randomized. In each case, we collect 20 tele-operated human demonstrations to pre-train the diffusion policy and the critic via Cal-QL + *PA-RL* that we then fine-tune online. For task **(b)**, we consider a "distribution shift" fine-tuning scenario, where the demonstrations show no distractors, but fine-tuning is done with distractor objects. While seemingly benign, this sort of difference between pre-training and fine-tuning setups is still challenging as it leads to poor fine-tuning performance (Kumar et al., 2022).

**Fine-tuning setup and comparisons.** In each case, we fine-tune with a sparse reward function that is based on the detected positions of the target objects and the gripper state. After every robot trial, we perform a manual reset and randomization of the object position and orientation. When running

*PA-RL* on the real robot, we found it important to collect 20 warmup episodes from the pre-trained policy before updating it. We also compare our approach to a filtered BC for autonomous improvement, based on Zhou et al. (but without goal conditioning or diffusion policy) for one of the tasks **(task (a))**. We omit this comparison for task **(b)** since the pre-trained DDPM policy did not produce any successes under distribution shift on task **(b)** for seeding iterative filtered BC. We found the diffusion policy to be brittle on task **(b)**.

Offline pre-trained initialization

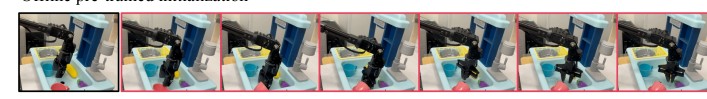

Common failure mode during online fine-tuning

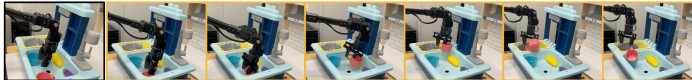

After 50 episodes of online fine-tuning

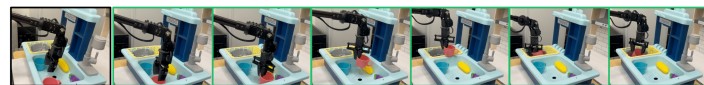

Figure 3: **Evolution of learned behaviors during online fine-tuning with *PA-RL* on task (a), with a new cup placement**. The offline initialization (in red) fails to both grasp the cup and place it on the rack. During intermediate online interaction episodes (in yellow), it successfully grasps the cup, but fails to place it on the rack. After 50 episodes (in green), it learns to successfully grasp the cup and place it on the rack.

**Real-robot fine-tuning results.** We observed significant and efficient performance improvement on both tasks when fine-tuning with *PA-RL*, resulting in a 20-35% higher success rate within 50-110 minutes. We noticed a performance drop during the first 50 episodes of fine-tuning in the *"cup to drying rack"* task, which was consistent with our findings in CALVIN task and many other works studying online fine-tuning (Nakamoto et al., 2024). Our policy enables the robot to quickly recover its behavior and show improvement within the next 20 episodes.

## 5.3 ABLATION STUDIES AND CONTROLLED EXPERIMENTS (APPENDIX D)

We ran some ablation experiments to understand the importance of each component of *PA-RL*. Concretely we aim to answer: **(1)** when is global optimization (Equation 4.1) critical for improving the policy? and **(2)** when is local optimization (Equation 4.2) critical for improving the policy? On the two tasks we study (antmaze-large-diverse and CALVIN), we make

| Task | *PA-RL* no global opt. | *PA-RL* no local opt. | *PA-RL* |
|---|---|---|---|
| antmaze-large-diverse | $0 \rightarrow 0$ | $74 \rightarrow 95$ | $73 \rightarrow 93$ |
| CALVIN | $215 \rightarrow 389$ | $201 \rightarrow 357$ | $234 \rightarrow 455$ |

Table 4: **Understanding the importance of global and local optimization.** We compare the performance of *PA-RL* + Cal-QL with and without global optimization as measured by average return obtained Note that not using both local and global optimization leads to worse performance. On diverse data such as antmaze-large-diverse, we find global optimization is crucial. On somewhat more narrow data, (e.g., play data in CALVIN) local optimization is also important.

a number of interesting observations. First, we find that both local and global optimization are critical for performance on some environment: on antmaze-large-diverse global optimization is critical, but local optimization is not as important. On CALVIN, both of the components are important. This tells us that global optimization is important in general, but local optimization is perhaps only useful when we have a somewhat narrow dataset (e.g., action coverage on CALVIN is narrow; while action coverage on antmaze is quite high). Thus, *we recommend the workflow* of always deploying global optimization when running *PA-RL* and strongly using local optimization when the dataset action distributions are somewhat narrow to make more targeted edits to the actions locally.

**Discussion and Conclcusion.** In this paper, we developed *PA-RL*, a method to fine-tune policies of various classes and parameterizations via actor-critic RL. We showed state-of-the-art online fine-tuning results across a number of simulation tasks and on two real-robot tasks. Despite promising results, *PA-RL* still has some limitations that future work should aim to address. Most importantly, *PA-RL* requires sampling multiple actions from the policy, which is expensive for large foundation policies. That said, future work can attempt to reduce this computational cost by caching actions from past rounds and training on them using ideas from off-policy policy gradient. Understanding interplay between global and local optimization better is also a viable direction.

# 6 REPRODUCIBILITY STATEMENT

In order to foster reproducibility of our work, we have outlined all the implementation details needed to implement our method in Appendix B.1 and Section 4. We have also provided more information about our experiments and settings in Appendix A and B.1 along with a listing of our hyperparameters. Code to reproduce our results will be made available upon acceptance of this paper.

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

# Appendices

## A    ENVIRONMENT DETAILS

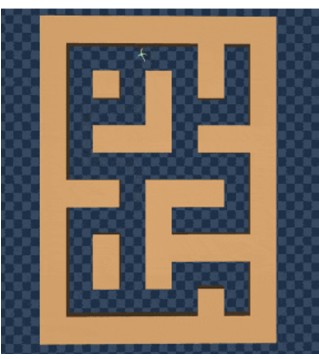
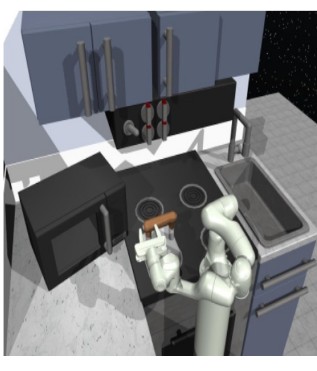
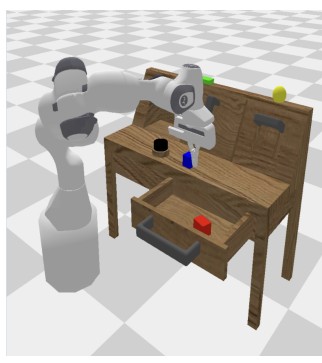

(a) Ant Maze Environment    (b) Franka Kitchen Environment    (c) Calvin Environment

Figure 4: Simulation Environments

**D4RL AntMaze:** We test methods across two maze sizes (medium and large) and two dataset types (play and diverse). The diverse and large datasets differ in the starting locations and goal locations of trajectories. The diverse dataset consists of trajectories with random initial and goal locations, whereas play contains a set of specific hand-picked locations. The offline datasets for this benchmark have high coverage over states and actions.

**D4RL FrankaKitchen:** The FrankaKitchen benchmark contains three tele-operated datasets: kitchen − complete, which contains trajectories that fully solve all sub-tasks, but is 37 times smaller than the other datasets; kitchen − partial, where there are both trajectories that fully solve all sub-tasks, and undirected data that performs unrelated behaviors; and kitchen − mixed, where no trajectory solves all tasks, requiring exploration from the agent.

**Calvin:** We use the task setup introduced by Shi et al. (2023), in which the robot arm needs to complete four tasks (OpenDrawer, TurnonLightbulb, MoveSliderLeft, and TurnonLED), with the distinction that we only use image observations (i.e., the agent doesn't have access to proprioception nor object states). To ensure Markovian rewards, we make the reward function is equal to the number of completed sub-tasks at each time-step (i.e., the agent only gets reward +4 if all sub-tasks are completed). The evaluation score for a trajectory is the maximum number of sub-tasks completed simultaneously at any single point in the trajectory.

Results for all environments and experiments are averaged over 5 random seeds and 32 evaluations per seed at each evaluation time-step (Figure 2). Scores are scaled from [0, 4] to [0, 100]. Shaded regions in the plots are standard errors over random seeds.

## B    EXPERIMENT DETAILS

### B.1    DETAILS AND HYPERPARAMETERS FOR *PA-RL*

**Action optimization hyperparameters:** For all experiments shown on the paper except for ablations, the number of actions sampled from the base policy is **32**, which are filtered down to the top **ten**, and then propagated through the Q-function for **ten** gradient steps with gradient step size of **3e-4**. While we find that these values are robust to all the tested settings, these choices might require changes according to the characteristics of the available dataset and action space. For example, larger action spaces (such as bimanual manipulation) might require larger gradient step sizes or close-to-optimal datasets might perform well with significantly fewer action samples and gradient steps.

**Distributional critic:** When any of the random seeds in a domain showed instability in the critic pre-training (i.e. had exploding Q-values) we switched the critic from an MLP that predicts the continuous action value to a distributional critic and trained with the HL-Gauss loss (Farebrother et al.) instead. Specifically, we switched to a distributional critic for the AntMaze and FrankaKitchen domains, and we trained with MSE on Calvin and the real robot experiments.

**Sampling vs argmax for action candidate selection:** For environments in which CQL/Cal-QL used the max-backup version of Q-target calculation (namely, all 4 AntMaze environments), we find that taking the argmax of $\pi_\phi^{\mathrm{Opt}}$ during inference yielded slightly faster convergence than sampling from the considered actions. During **policy distillation**, to decide whether to imitate only the argmax of $\pi_\phi^{\mathrm{Opt}}$ or whether to imitate all samples, we keep track of the variance of action candidate Q-values during pre-training. If the variance is too small, we find that training only with the argmax performs better. Otherwise, training with samples from the categorical distribution yields slightly better results.

| Environment | Policy Training Argmax Action | Policy Training Softmax |
|---|---|---|
| kitchen-partial-v2 | 89.375 | 95.3125 |
| kitchen-complete-v2 | 90.3125 | 94.53125 |
| kitchen-mixed-v2 | 67.96875 | 75.15625 |
| CALVIN | 60.6771 | 46.5625 |

Table 5: **Comparison between doing policy distillation with samples from $\pi_\phi^{\mathrm{Opt}}$ and only the argmax.**

| Environment | STD of Action Candidate Q-values |
|---|---|
| kitchen-partial-v2 | 1.56 |
| kitchen-complete-v2 | 2.66 |
| kitchen-mixed-v2 | 11.54 |
| CALVIN | 0.02 |

Table 6: **Standard deviation of the Q-values of action candidates ($\widetilde{\mathcal{A}}_{\pi,m}^T$) during pre-training.**

**Details for image-based domains:** Following Yarats et al. we augment image observations with random shift augmentations of 4 pixels. To mitigate the failure case in which the Q-values for different actions on the same state collapse to the same value, we use the Q-function architecture introduced by Kumar et al. (2022). At every layer of the critic MLP, we concatenate the action vector to the inputs, so that the network places more importance to the actions.

**Base policy hyperparameters:** We use the same Diffusion Policy architecture and training hyperparameters as IDQL (Hansen-Estruch et al., 2023). In particular, we use batch size 1024, T=5 diffusion steps, cosine beta schedule, the LN_Resnet architecture with hidden dimension size = 256 and n = 3 blocks. We pre-train the diffusion policy with learning rate decay but with a constant learning rate during fine-tuning. For image-based domains (CALVIN and real robot) we use a ResNet 18 encoder trained from scratch. For the auto-regressive transformer policy, we discretize each action dimension into 128 bins, and do not use discretization for the state observations. We use a transformer architecture with 4 layers, 256 hidden size, 8 heads, and learning rate 3e-5.

**Reward scale and bias:** To maintain consistency of hyperparameters across all domains, we bias all rewards from the offline dataset and replay buffer such that the maximum possible timestep reward is zero, and other possible rewards are negative. In particular, we use bias = -1 for AntMaze and real robot, and -4 for FrankaKitchen and CALVIN.

**Cal-QL hyperparameters:** We carry over most hyper-parameter choices from Cal-QL: critic architecture and learning rate, discount, mixing ratio.

**Table of hyperparameters:**

| | |
|---|---|
| **Critic LR** | 3e-4 |
| **Discount $\gamma$** | 0.99 |
| **Critic batch size** | 256 |
| **Base policy batch size** | 1024 (Diffusion Policies), 256 (Transformers) |
| **CQL $\alpha$** | 0.005 (AntMaze), 0.005 (Kitchen), 0.01 (CALVIN & Real robot) |
| **Mixing ratio** | 0.25 (Kitchen), 0.5 (Rest) |
| **Optimizer (critic and base policy)** | Adam (Kingma & Ba, 2015) |
| **Critic pre-training grad steps** | 1e6 (AntMaze), Rest: 5e5 |
| **Base policy grad steps** | Diffusion policies: 3e6 
 Transformers: 2e6 |
| **Critic hidden layer sizes** | [256, 256, 256, 256] (AntMaze), [512, 512, 512] (Rest) |

### B.2 DETAILS AND HYPERPARAMETERS FOR BASELINES

**IDQL** We use the IDQL-Imp version of IDQL, in which the Q-function, the value function, and the diffusion policy are fine-tuned with new experiences. We use the same network architectures as *PA-RL*. For the IQL $\tau$ expectile, we use 0.9 for AntMaze and 0.7 for everything else. We remark that results for IDQL are not entirely comparable to their paper because Hansen-Estruch et al. (2023) used the "-v0" antmaze datasets from D4RL, but Fu et al. (2020) deprecated the "-v0" datasets in favor of "-v2" due to a bug associated with termination flags in -v0 datasets.

**DQL** We extensively tuned DQL for fine-tuning in the absence of any official fine-tuning results. For the main $\eta$ RL weight hyperparameter, we performed an environment-specific hyperparameter search at the pre-training phase, selected the one that performed best, and then kept $\eta$ fixed for fine-tuning. For AntMaze tasks we tried $\eta = \{0.05, 0.5, 1, 3, 3.5, 5, 7, 9, 11, 13, 15\}$. We chose $\eta = 11$ for large-diverse, $\eta = 15$ for large-play, $\eta = 9$ for medium-diverse, and $\eta = 7$ for medium-play. For FrankaKitchen tasks we tried $\eta = \{0.005, 0.01, 0.05, 0.1\}$. For partial, complete, and mixed, we chose $\eta = 0.005$. For CALVIN we tried $\eta = \{0.01, 0.1, 1, 5, 10, 15\}$. We picked $\eta = 0.01$. For offline checkpoint selection, we follow the original methodology of selecting the checkpoint with second lowest DDPM loss, saving checkpoints every 50k gradient steps.

**Cal-QL** Since we branch off our hyperparameter choices from Cal-QL, this baseline shares most of *PA-RL*'s hyperparameters. We used (256, 256) hidden sizes for the policy architecture for every environment.

**DPPO** We train a diffusion-based PPO policy based on a DPPM model pretrained on an offline dataset in each simulated task. For the state-based tasks AntMaze and FrankaKitchen, we train DPPO-MLP with 40 parallelized environments and an action chunking size of 6 for AntMaze and 8 for FrankaKitchen. For the pixel-based task CALVIN, we train DPPO-ViT-MLP with 50 parallelized environments and an action chunking size of 4.

**RLPD** For Table 2, we train a gaussian policy from scratch with UTD ratio of 10 (same as with Diffusion *PA-RL* + RLPD), critic ensemble size ten, and critic ensemble subsample size of two.

## C    REAL-WORLD FINE-TUNING OF OPENVLA WITH *PA-RL*

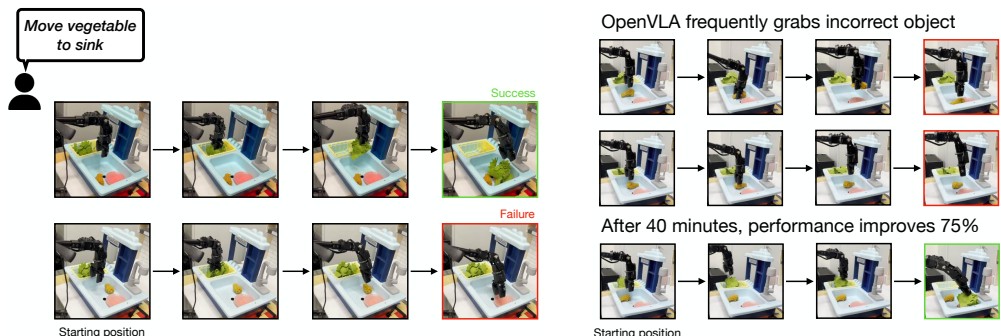

(a) Zero-shot language-based trials with OpenVLA      (b) Online fine-tuning with *PA-RL*

Figure 5: **Filmstrips of the manipulation task we fine-tune OpenVLA on.** (Left) the new task, "vegetable to sink", requires identifying the vegetable from the distractor (a fried chicken wing), grasping it, and placing it on the pink plate. We collect 50 trials by zero-shot prompting OpenVLA to solve the task. 40% of the trials are successful. (Right) we deploy *PA-RL* to improve OpenVLA for this task, interacting on the real-robot. We observe that OpenVLA frequently grasps the distractor object instead of the vegetable. After 40 minutes of wall clock time, we evaluate the resulting fine-tuned policy. OpenVLA + *PA-RL* attained a 70% success rate.

We present a real-world fine-tuning result of OpenVLA Kim et al. (2024), the 7B-parameter generalist robot policy. *PA-RL* improves OpenVLA performance by 75% on a real-world manipulation task after 1 hour of zero-shot language-conditioned trials, and 40 minutes of online RL fine-tuning on the real robot.

**Task and experimental setup:** We consider a new task in the same kitchen environment as our previous two real-world tasks: "vegetable to sink", which requires grasping a toy cabbage and placing it on a plate in the sink. There additionally is a distractor on the scene. We collect 50 rollout episodes by zero-shot prompting OpenVLA with the instruction "put the vegetable on the plate", and use them to pre-train a Q-function with Cal-QL + *PA-RL*. Note that while our toy kitchen resembles a kitchen that was present in the training dataset for OpenVLA, the specific task is novel and there are likely significant differences in camera angles and background that affect OpenVLA zero-shot performance. The base OpenVLA model achieves a 40% success rate on this task.

**Results:** After pre-training the critic, we run PA-RL + Cal-QL fine-tuning in the real world for 40 minutes (which includes both robot interaction time and OpenVLA training time) with a sparse reward function, manual resets of the environment, and object randomization, similarly to the previous real-robot experiments. The resulting fine-tuned OpenVLA policy obtained a 70% success rate, which is 75% higher than the base OpenVLA, and 40% higher than without the 40 minutes of real-world fine-tuning (i.e., offline only). **We believe that this is the first result that fine-tunes a large generalist policy with 7B parameters with actor-critic RL successfully in the real world.**

**Systems and implementation details for OpenVLA:** To accelerate training, after each epoch of policy training we maintain a cache to store actions the fine-tuned OpenVLA policy would take at each state by sampling 16 actions from this generalist policy. This cache enables the Q-function training in Cal-QL to still run at similar speeds as it would have with a much smaller policy, because actions in the cache can be utilized for TD backups for multiple gradient steps. To speed up action caching, we ran 12 distributed processes to cache OpenVLA actions after each epoch of training. Since the pre-training stage doesn't update the base policy parameters (distillation only comes in during fine-tuning) we only need to cache at the beginning of that stage. During online fine-tuning, we now update the parameters of the generalist OpenVLA policy. Concretely, we distill optimized actions into OpenVLA via LoRA fine-tuning with rank=32 to speed up training. During environment interaction, we also run action optimization at inference. In this case, we reduce the the number of action samples used for a single observation from OpenVLA to 4 to be able to maintain an action frequency of 3hz. Aside from reducing the number of samples from the base policy due to memory constraints, and reduced distillation learning rate for stability, all hyperparameters are the same as used to fine-tune diffusion policies.

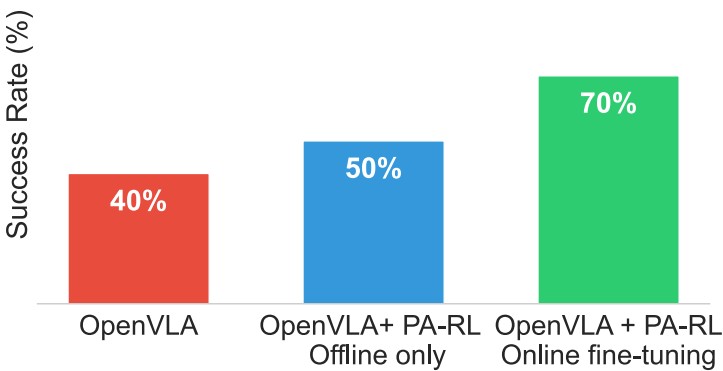

Figure 6: **OpenVLA real-robot fine-tuning results.**

# D    ADDITIONAL FIGURES

## D.1    REAL ROBOT FINE-TUNING ON TASK (B)

Offline pre-trained initialization

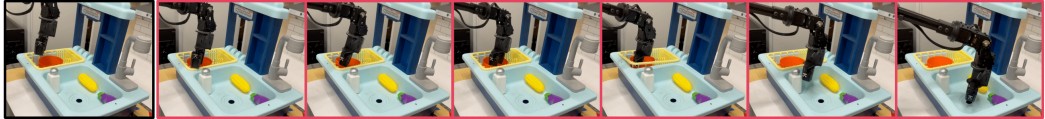

After 10 episodes of online fine-tuning

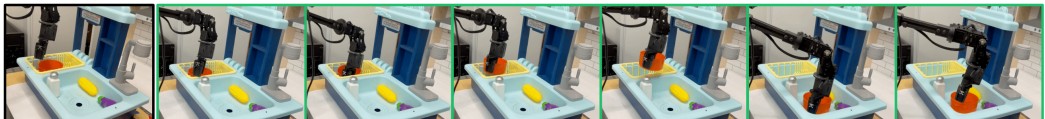

Figure 7: **Evolution of learned behaviors during autonomous online finetuning of *PA-RL* on task (b) on a difficult pot placement**. The offline initialization (in red) fails to grasp the pot, and gets stuck when attempting to move it to the sink. After only 10 online fine-tuning episodes (in green), *PA-RL* learns to successfully complete the task.

## D.2    LEARNING CURVES FOR AUTO-REGRESSIVE TRANSFORMERS AND IQL WITH *PA-RL*

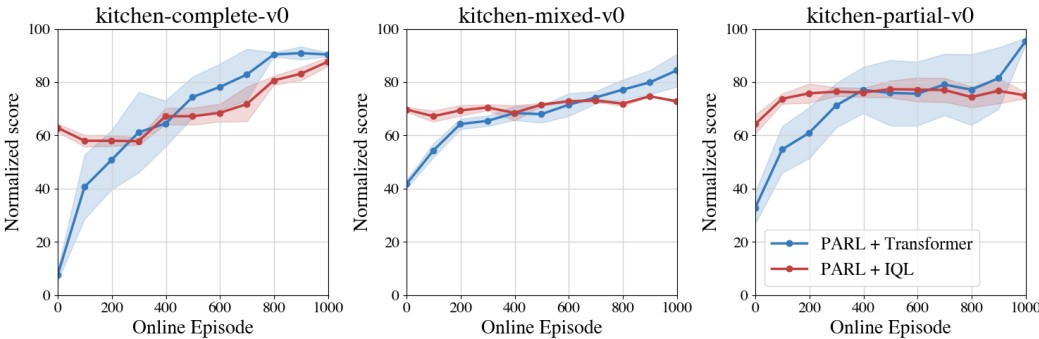

Figure 8: **Learning Curves for Auto-Regressive Transformers with *PA-RL* and Cal-QL, and Diffusion Policies with *PA-RL* and IQL.**

## D.3 LOCAL AND GLOBAL OPTIMIZATION ABLATION EXPERIMENTS

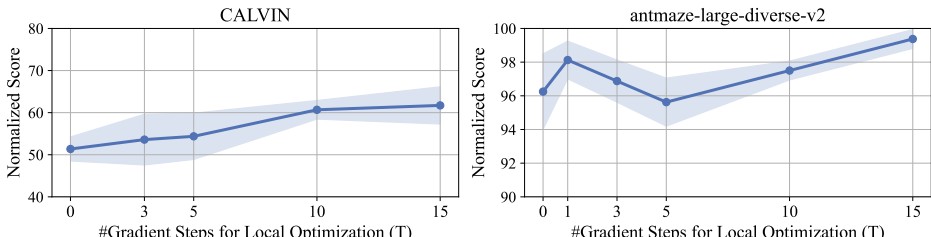

Figure 9: **Ablation for the number of gradient steps for local optimization (T).** We plot the evaluation performance for *PA-RL* + Diffusion Policy at the end of a fine-tuning budget of 1k episodes on CALVIN (left) and antmaze-large-diverse-v2 (right), taking different numbers of gradient steps during the Local Optimization procedure. We chose to analyze the effect of local optimization on these two tasks because they sit on opposite sides of the data coverage spectrum: CALVIN features relatively little coverage over actions, since the provided dataset is "play data", while antmaze-large-diverse-v2 provides high-coverage over actions (as measured by delta x, delta y, which is more relevant to the task). (Left) CALVIN benefits significantly from increased number of gradient steps, getting up to 20% increase in final performance compared to taking no gradient steps. (Right) antmaze-large-diverse-v2 already reaches 96% success rate without taking any gradient steps (i.e., without the local optimization step). We hypothesize that because of the high-coverage, using global optimization with a large-enough number of samples from the base policy already recovers good actions.

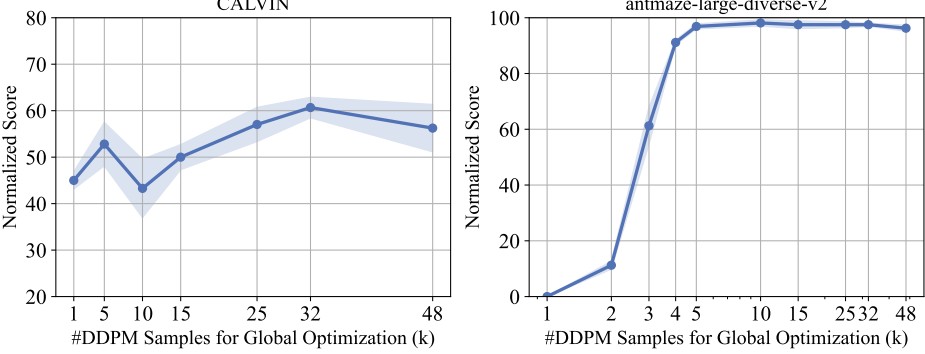

Figure 10: **Ablation for the number of samples from the base policy (k).** We plot the evaluation performance for *PA-RL* + Diffusion Policy at the end of a fine-tuning budget of 1k episodes on CALVIN (left) and antmaze-large-diverse-v2 (right), sampling different number of actions from the base policy to generate action candidates both for policy distillation and during inference. (Left) CALVIN benefits significantly from increased number of samples from the base policy, attaining 33% higher normalized score when taking 32 samples (the default value used for *PA-RL*) from the policy compared to only 1 sample. (Right) antmaze-large-diverse-v2 exhibits a sharp decrease in final performance when taking fewer than 5 samples from the base policy.

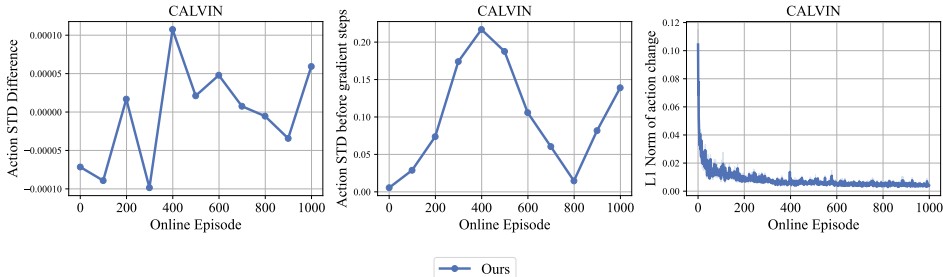

Figure 11: **Analysis of the effects of local optimization.** To test whether local optimization results in duplicated action samples, we plot the difference between the standard deviation of action samples before and after taking gradient steps (left) during evaluation episodes on the CALVIN task throughout fine-tuning. The difference in standard deviations is extremely low throughout training. Further, to ensure action samples were not largely duplicates to begin with, and to put the value scale into perspective, we plot the raw standard deviation of action samples before taking gradient steps (center). Standard deviation of actions changes by less than 0.1% on average during training. Thus, local optimization does not lead to action sample duplication. (Right) we plot the L1-Norm of the change in actions by the local optimization procedure (i.e. the L1 norm of the difference in actions before and after the gradient steps). The biggest direct effect on actions happens in the beginning of fine-tuning, and it quickly decays throughout online training. Note that because of policy distillation, action changes from the local optimization step are compounding (i.e., the actions before applying the gradient steps have already been optimized in past iterations). This might explain the decay in action changes from local optimization.

## D.4 CEM OPTIMIZER + RANDOM INITIALIZATION COMPARISONS

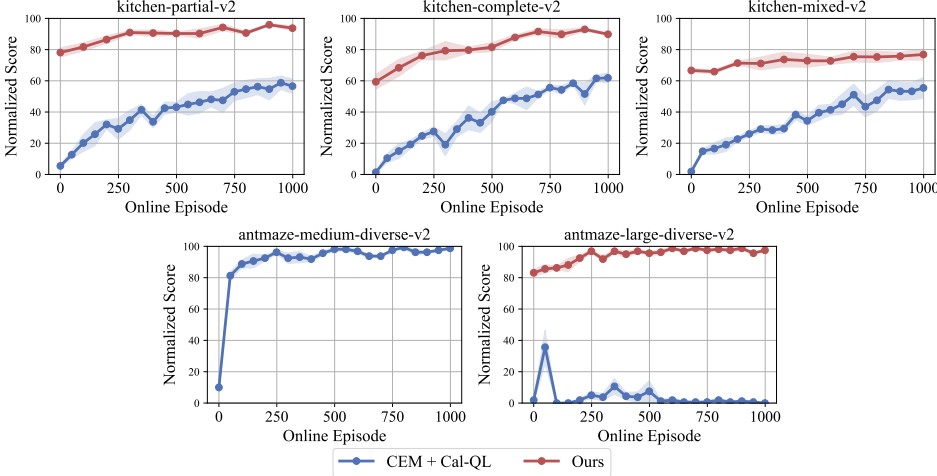

Figure 12: **Comparison with CEM optimizer.** Instead of using the action optimization procedure detailed in Section 4, any time the Cal-QL algorithm queries the policy we perform a Cross-Entropy Method optimization process to obtain actions. We use the same CEM hyper-parameters as Simmons-Edler et al. (2019), and maintain the Cal-QL hyper-parameters and architectures as *PA-RL*. for all tested environments, the performance after pre-training (i.e. at step 0, before taking any online steps) is at or close to 0, and performance improves over the course of fine-tuning, but remaining well below PA-RL with a diffusion policy.

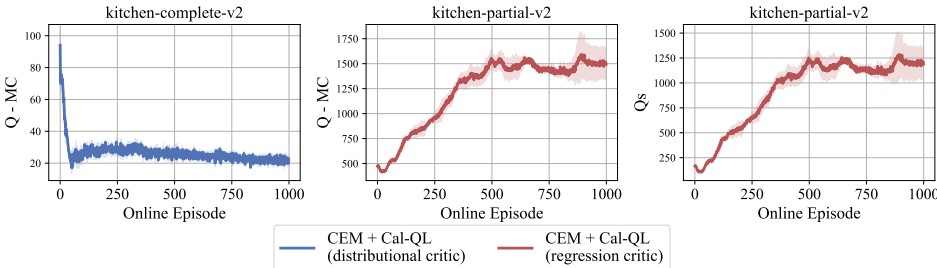

Figure 13: **CEM exploits Q-function over-optimism.** (Left) We plot the difference between predicted Q-values of CEM actions, and the Monte-Carlo discounted returns that those actions actually got, on kitchen-complete-v2, a task whose dataset contains optimal actions. The critic is trained in the same manner as in Figure 12. We observe that at the beginning of fine-tuning, predicted Q-values are much higher than the MC returns, even much higher than the predicted Q-values further into training, when task performance is much higher (see Figure 12). This points to the fact that the CEM optimizer is able to find actions that maximize the Q-function, but are not actually good. (Center) We repeat the same experiment but with a regression-trained critic instead of a distributional critic trained with HL-Gauss. The distributional critic bounds the predicted values by design, which limits over-estimation. By training a Cal-QL critic without a fixed value range (on kitchen-partial-v2), we see much larger over-estimation of Q-values. In fact, predicted Q-values become large positive numbers (right), where rewards for this task are always non-positive.

### D.5    CEM OPTIMIZER + PRE-TRAINED POLICY INITIALIZATION

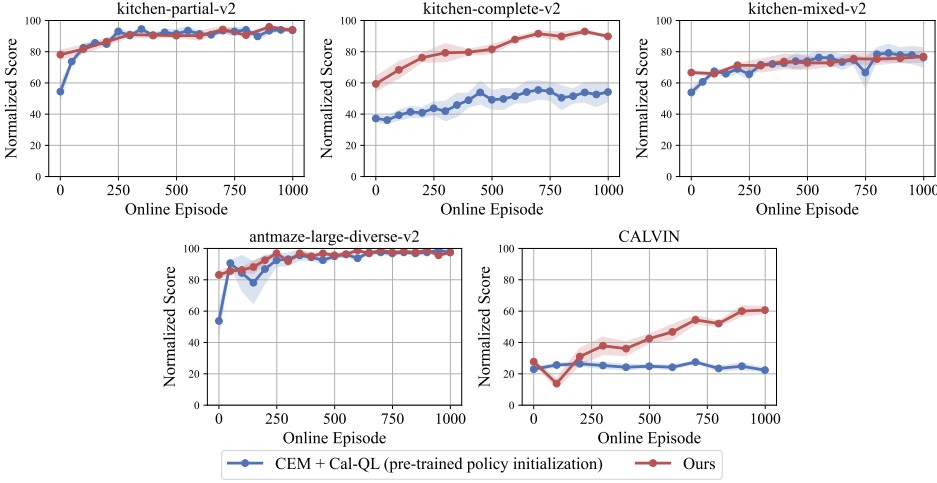

Figure 14: **Comparison with CEM optimizer with a pre-trained policy initialization.** We compare to using a CEM optimization procedure where the initial population of actions comes from the same pre-trained policy used for *PA-RL*. *PA-RL* results in 42% better offline-only performance across tested domains. In antmaze-large-diverse-v2, kitchen-partial-v2, and kitchen-mixed-v2, CEM quickly catches up and ends with very similar asymptotic performance. In kitchen-mixed-v2 and CALVIN *PA-RL* significantly outperforms CEM, with 66% and 172% better performance respectively. kitchen-complete-v2 and CALVIN have lower coverage of actions in their datasets, and CALVIN has highly multi-modal data. We hypothesize these dataset characteristics, which are highly common in real-world robotics datasets, are hurting CEM performance, since CEM can average the different modes of behavior, resulting in OOD actions. Further, CEM lacks an equivalent of the local optimization step to direct exploration towards actions the critic rates highly.

### D.6 COMPARISON WITH SELF-IMITATION LEARNING

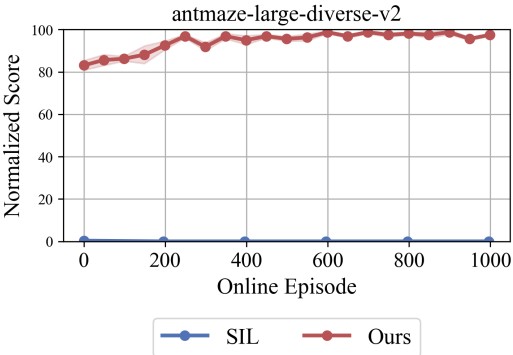

Figure 15: **Comparison with Self-Imitation Learning (Oh et al., 2018) on antmaze-large-diverse-v2.** We implement a Diffusion-Policy version of Self-Imitation learning (i.e., single action sample from the replay buffer weighted by positive advantages) on top of our codebase by disabling local optimization and global optimization (i.e., sampling a single action from the base policy, and not taking any gradient steps), and adding an exponentiated advantage weight to the policy distillation targets. For fairness, critic pre-training and fine-tuning are done in the same manner as *PA-RL*. On antmaze-large-v2, Self-Imitation Learning never attains positive performance. We hypothesize that poor performance is due to taking a single sample from the base policy (Figure 10 shows that taking more samples greatly improves performance).

### D.7 COMPARISON WITH COMPUTING ACTIONS FOR BELLMAN BACKUP WITH THE BASE POLICY

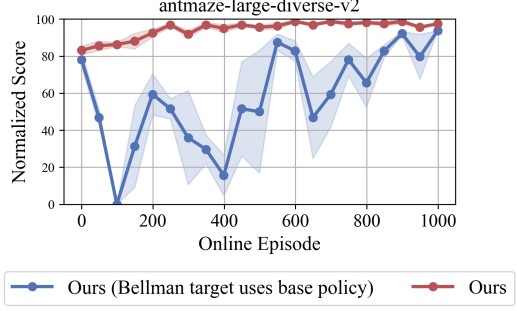

Figure 16: **Ablation for the choice of using the optimized action for Bellman backups.** To ablate the choice of computing targets using the optimized policy $\pi^{\mathrm{Opt}}_{(\phi,\theta)}(\cdot|\cdot, m)$, we compare it against directly sampling from the base policy $\pi_\phi$, and test it on antmaze-large-diverse-v2 fine-tuning. Both methods start from the same pre-trained critic checkpoints. Using the base policy for Bellman targets makes fine-tuning much more unstable, with a sharp drop in performance in the beginning, but ultimately obtains similar performance.

## D.8 LEARNING CURVES FOR GAUSSIAN POLICIES WITH PA-RL

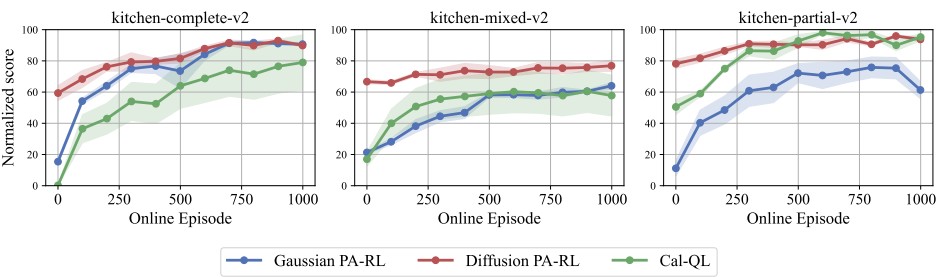

Figure 17: **Learning curves for gaussian policies with *PA-RL*, compared with Diffusion Policies with *PA-RL* and the standard Cal-QL with gaussian policies.** As with other experiments, we first train the base gaussian policy with BC on each dataset, and then do critic pre-training, followed by online RL fine-tuning. The only hyper-parameter we change for gaussian policies is the distillation learning rate, setting it to 3e-4. We observe Gaussian *PA-RL* performs competitively with the standard Cal-QL on kitchen tasks.

## E TRAINING TIME DISCUSSION

*PA-RL* optimizes actions using the procedure described in Section 4 any time an action from the policy is needed. We discuss how this affects the App of our method at different stages.

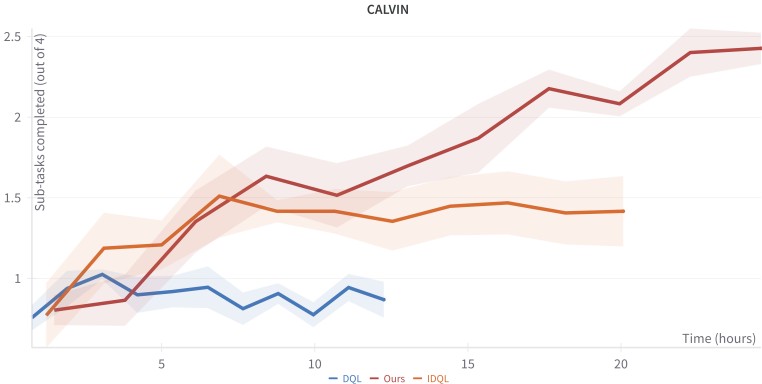

Figure 18: **Performance on CALVIN task as a function of wall clock time for *PA-RL*, IDQL, and DQL.** All three methods ran on the same compute instance type (TPU v4), were implemented in the same codebase. Observe that *PA-RL* improves at a similar rate per unit amount of wall-clock time as IDQL, but is able to improve far beyond to a better performance value. DQL largely remains flat as a function of more unit wall-clock time put into training.

**Critic training.** In principle, action optimization should increase memory and computation requirements to critic training, but it also enables using an action cache to compute ahead of time, even in a distributed manner, when sufficient numbers of actions from the base policy are available. To make sure that this cache is not stale and to ensure that the critic models the optimal / on-policy value function, the actions cache is updated after every epoch of policy training via supervised learning. When sampling from the base policy is more than T times more expensive than taking T gradient steps of the critic (as is the case with OpenVLA or with diffusion policies with a large number of denoising steps), *PA-RL* can be significantly more efficient than alternatives that do not do caching.

**Policy distillation.** Compared to standard offline RL and online fine-tuning objectives, the supervised learning objective *PA-RL* can be significantly more efficient than policy improvement through reparameterization. For example, for a diffusion policy, backpropagating critic gradients through the

diffusion chain uses a larger memory footprint than the DDPM objective *PA-RL* uses, by a factor equal to the number of denoising steps.

**Inference.** During inference, *PA-RL* can optionally also apply action optimization by querying the base policy multiple times to sample an action. This can significantly increase the memory requirements of our method. That said, we do note that the number of samples from the base policy during inference can be much smaller than during training, as we do with OpenVLA (see Appendix C). *PA-RL* additionally requires taking multiple gradient steps of the critic with respect to the actions. We note that depending on the architecture used, this can be much cheaper than doing multiple full forward passes through the Q-function. For example, for image-based domains, the bulk of the computation happens for image encoding, which does not depend on the action. Therefore, the gradient steps will ignore that part of the network. There is also room for improvement for future work to investigate reducing the number of gradient steps further into training (as Figure 11 right suggests local optimization might have diminishing effects as fine-tuning progresses).

