# OpenReview forum: "Parameterization Agnostic RL"
_ICLR.cc/2025/Conference — Submitted to ICLR 2025_

### Official Review · Reviewer_Eej1 · 2024-10-16

**Soundness:** 3
**Presentation:** 2
**Contribution:** 3
**Rating:** 5
**Confidence:** 3

**Summary:**

The authors propose an approach to Reinforcement Learning that is agnostic to the parameterization of the policy. The proposed framework can seamlessly work to train diffusion policies and transformer policies.

**Strengths:**

1. The paper investigates an important topic in RL.

2. The presented algorithm performs strongly against several baselines on different offline, online fine-tuning, and hybrid RL tasks, including two real-world experiments.

**Weaknesses:**

~1. How should the presented method be modified to work with unimodal policies like Gaussian policies? Currently, the presented method seems irrelevant for unimodal policies, as the optimized actions would converge to the same point. This assumption is not mentioned in the paper and makes the proposed method less RL-agnostic.~

~2. While the proposed method achieves high returns in the presented experiments, a discussion on the training time is missing. Plotting PA-RL's performances along with the baseline's performances as a function of the clock time for the offline experiments would increase the significance of the submission.~

3. Many mistakes and issues alter the reading flow of the paper (see Remarks below).

4. The code is not provided, which limits the reliability of the experiments.

3. Some design choices are not justified (see Questions below).

~6. An analysis of the distribution of the actions after the local optimization would be interesting as there is a risk that the distribution of the actions would not be multi-modal after the optimization, resulting in duplicated samples. As this property depends on the critic’s behavior, analyzing the critic’s landscape w.r.t. the actions would also be beneficial.~

**Questions:**

~1. The influence of the number of gradient steps $T$ performed for the local optimization is not discussed.~

~2. The appendix mentions that the critic of PA-RL was trained with a categorical loss for some experiments. Was the same mechanism applied to the baseline as well?~

3. Line 138, why did you choose to add the $\frac{1}{1 - \gamma}$ in the definition of $V^{\pi}$? The expectation over the transition dynamics is missing.

4. ~In Line 6 of Algorithm $1$, the authors verify that the locally optimized action yields a higher value when evaluated on the critic. Is this step needed? This operation adds a computing cost that is not justified in the paper.~ Moreover, in its current version, nothing is done to change the optimization process when the new action is worse than the previous one, which means that the following iteration of the for loop will be identical.

~5. In Line 4 of Algorithm $2$, the Bellman backup is computed from the optimized action $\tilde{\mathcal{A}}_{\pi, m}$ instead of the action generated from the current policy. The authors do not show the benefit of this choice.~

– Remarks –

~A. Line 102, “compare EMaQ” should be “compared to EMaQ”.~

~B. Line 121, “self-tunign” should be “self-tuning”.~

~C. Line 137, the optimal policy attains the maximal cumulative discounted “reward” and not “value function”.~

D. Most abbreviations used in the paper are not defined, which makes the paper less accessible to a broader community. For example, nearly all baseline abbreviations are not explained, and the abbreviation AWR is not explained either.

~E. Equation $3.1$, $\bar{\alpha}$ is not defined.~

~F. Line 156, the definition of the action “a” is wrong. The action “a” is described as a sum of independent terms, while the reverse diffusion process is an iterative process where each term depends on the previous one.~

~G. Equation $3.3$, the $\alpha$ parameter plays no role in the minimization. Additionally, the minimization is not performed on the conservative regularizer alone. Finally, $\pi$ is only defined $2$ sentences after the equation is presented, which can be misleading.~

~H. Line 194, a space is missing in “RLalgorithm” between “RL” and “algorithm”.~

~I. Line 228, “maximize rewards” should be replaced by “maximize returns”.~

~J. Line 237, it might be beneficial to replace $\pi$ by $\phi$ in $\tilde{\mathcal{A}}_{\pi, m}$, to stress its dependency in the parameters of the learned policy.~

~K. In Line 6 of Algorithm $1$, the right side of the inequation should be $Q(s, a^{(i-1)})$ instead of $Q(s, a^{(i)})$.~

~L. In Line 2 of Algorithm $1$, I would replace “at every state” with “for every state s”.~

~M. Equation 4.5, I would add the dependency on $\theta$ as an input to the loss.~

N. In Line 4 of Algorithm $2$, the loss is briefly explained in Section $3$ but not properly defined. Writing its definition would help.

O. In Line 5 of Algorithm $2$, the loss is not defined, making the pseudo-code harder to read. ~Additionally, the loss should take as input $\theta$ as it also relies on it.~

~P. Line 323, “a a set” can be replaced by “a set”.~

Q. I found the last paragraph of Section $4.3$ unconvincing and unclear. Reformulating it more straightforwardly would help clarify the authors' argument. For example, the terms in Equation $4.6$ could be introduced before the equation is presented, and then its significance could be explained.

~R. Line 353, “that train diffusion policies” could be replaced by “that also train diffusion policies”. This would make the sentence clearer.~

S. The authors only report the archive versions of the paper, while a significant proportion of the cited works are peer-reviewed. Reporting the venue of the works would be more accurate.

~T. The paper Li, Zechu, et al. "Learning Multimodal Behaviors from Scratch with Diffusion Policy Gradient." NeurIPS (2024) is close to the presented work. It would be worth including it in the Related Work.~

---

> ### Author Response · Authors · 2024-11-19
>
> We sincerely thank the reviewer for the very detailed and constructive feedback. We found your feedback to be very helpful and have updated the paper to address all the main writing concerns raised. We have added new experiments to understand the effect of local optimization steps, global optimization steps, scale of the policy (as we discuss in the global comment, PA-RL can successfully improve the success rate of OpenVLA, a 7 billion parameter generalist policy, on real-world tasks by 75%), added experiments on PA-RL + Gaussian policies, and added an analysis of wall-clock time for PA-RL. We are also working on cleaning up the code and will work towards releasing a preliminary version by the end of the rebuttal period, and a version accompanying the final paper.  **Please let us know if your concerns are addressed, and if so, we would be grateful if you are willing to raise your score.** We would be happy to engage in a discussion with you.
>
> > a discussion on the training time is missing
>
> We added a new result **Figure 17** to the paper, in which we plot performance as a function of wall clock time for our method and two baselines (IDQL and DQL) (https://imgur.com/Cp9NGuG). All runs for these methods on CALVIN are trained on the same machine type (TPU-v4), and are implemented on the same codebase. TPU instances were used only for image-based domains, and experiments for other environments ran on a cluster with different kinds of GPUs, so they are not easily comparable.
> We have additionally added to **Appendix E** a discussion about the computational complexity of our method. In short, PA-RL significantly increases the memory requirements for inference, since it requires multiple samples from the base policy, but makes training with much larger policies more efficient by allowing action caching, and can be significantly more memory efficient for diffusion policies that employ a large number of denoising steps than alternative RL algorithms that improve the policy through reparameterization.
>
> > The influence of the number of gradient steps T performed for the local optimization is not discussed.
>
> We have now added a new experimental result in **Figure 9** to the paper to show an ablation for the number of gradient steps for local optimization (https://imgur.com/bm8w3Vk). We analyze two environments that sit on opposite sides of the data coverage spectrum: CALVIN, which features relatively little coverage over actions, since the provided dataset is “play data”; and antmaze-large-diverse-v2, which consists of high-coverage actions (i.e., coverage as measured by delta x, delta y, which is more relevant to the task).
>
> We find that CALVIN benefits significantly from increased gradient steps in local optimization, getting up to 20% increase in performance after fine-tuning compared to no gradient steps. On Antmaze, performance already reaches close to 100% even without any gradient steps. We hypothesize that because of the high-coverage, using global optimization with a large-enough number of samples from the base policy already recovers good actions (See **Figure 10** for an additional ablation for the number of samples from the base policy, which shows sampling at least 5 actions is critical to have strong final performance on antmaze-large-diverse-v2).
>
> Hence, the local optimization is important, especially in real-world robotic datasets that are typically collected via tele-operation as a result of which they exhibit smaller coverage.
>
> > How should the presented method be modified to work with unimodal policies like Gaussian policies?
>
> We have added an experiment in which we fine-tune Gaussian policies with PA-RL on the Kitchen tasks, obtaining comparable performance to Cal-QL (see **Figure 16**). This required no modification to the method, except changing the learning rate for policy distillation to 3e-4. In aggregate, we find that it performs similarly to Gaussian Cal-QL with reparameterization.
>
> | Task                |  Cal-QL | PA-RL + Diffusion Policy | PA-RL + Gaussian Policy |
> |---------------------|:-------:|:------------------------:|:-----------------------:|
> | kitchen-complete-v2 | 19 → 57 | 59 → 90                  | 15 → 91                 |
> | kitchen-mixed-v2    | 37 → 72 | 67 → 77                  | 21 → 64                 |
> | kitchen-partial-v2  | 59 → 84 | 78 → 94                  | 16 → 61                 |

---

> > ### Author Response · Authors · 2024-11-19
> >
> > > there is a risk that the distribution of the actions would not be multi-modal after the optimization, resulting in duplicated samples
> >
> > To understand whether local optimization results in duplicated samples, we plot both the standard deviation of action candidates before the local optimization step, and the change in standard deviation of actions produced by local optimization (see **Figure 11**, https://imgur.com/MC4eHRy). We observe that action standard deviations are relatively high throughout training (0.1 on average, when the actions have values between -1 and 1), and that throughout training local optimization almost does not change action standard deviations. This implies that the local optimization procedure is likely not resulting in duplicate samples, since action spread is maintained. While verifying multi-modality in large action spaces is statistically hard, the fact that we see significant improvements from using diffusion policies (over Gaussian policies) likely points to PA-RL benefiting from multi-modality of the policy class as well.
> >
> > > Was the same mechanism applied to the baseline as well?
> >
> > Yes, for fairness both baselines and our method use the same critic training objective and architecture.
> >
> > > the authors verify that the locally optimized action yields a higher value when evaluated on the critic. Is this step needed?
> >
> > Since we use a fixed and constant step size for local optimization for all domains and throughout training to reduce excessive hyperparameter tuning burden, there is a risk of taking too large of a step such that the Q-value of the resulting action actually decreases (i.e., a form of “overshooting”). In our preliminary experiments, we found that the number of gradient steps before hitting the overshooting condition is task-dependent: in real-world robot tasks we rarely ever hit overshooting either during critic training or during fine-tuning. On the other hand, for antmaze-large-diverse-v2, where local optimization has a much smaller effect (see **Figure 9**), most action optimization instances overshoot having taken between 1 and 5 gradient steps. Therefore, we added this condition to avoid overshooting.
> >
> > > This operation adds a computing cost that is not justified in the paper.
> >
> > We use jax.value_and_grad to efficiently compute both the gradient of the Q-function and the current action values at the same time. We then maintain the action with the highest Q-value seen so far with a batched where operation. This doesn't change the time complexity of the algorithm, and the added computation cost should be negligible.
> >
> >
> > > nothing is done to change the optimization process when the new action is worse than the previous one, which means that the following iteration of the for loop will be identical
> >
> > We implement local optimization in a batched way with XLA compilation on jax, which requires the shapes of tensors to be identical across loops. Since some actions in the batch might not have overshot, it is more efficient to keep running the optimization loop with all actions, even if this introduces empty loops, until we hit the termination condition based on the number of gradient steps.
> >
> > > the Bellman backup is computed from the optimized action $\tilde{\mathcal{A}}_{\pi, m}$ instead of the action generated from the current policy. The authors do not show the benefit of this choice.
> >
> > We have added a new ablation experiment for this choice of using optimized actions for computing TD targets. Please refer to **Figure 15**, which compares using the optimized policy $\pi^\mathrm{Opt}(\cdot|\cdot, m)$ and the base policy for computing next actions for Bellman backups on antmaze-large-diverse-v2 (https://imgur.com/3Yq312E). Using the base policy instead of the optimized policy leads to a sharp drop in performance in the beginning of training, and unstable fine-tuning during the allotted sample complexity budget, although it is able to eventually reach similar final performance as our method. In contrast, PA-RL attains a small cumulative regret and is able to quickly reach near 100% final performance while exhibiting stable fine-tuning trends.

---

> > > ### Author Response · Authors · 2024-11-21
> > >
> > > Dear reviewer Eej1,
> > >
> > > We wanted to check in on whether our responses, updates on the paper, new ablations and baselines addressed your concerns, and whether you had a chance to see the OpenVLA experiment we added. We would be happy to discuss further.
> > >
> > > Thank you!

---

> > > > ### Comment · Reviewer_Eej1 · 2024-11-23
> > > >
> > > > I thank the Authors for the thorough Rebuttal. I appreciate the effort made to tackle the concerns raised during the initial round of reviews.
> > > >
> > > > The requested ablations, the newly added experiment in Appendix C, and the reformulation of many sentences/paragraphs make the content of the updated submission largely different from the original submission. This is why I will keep my score and argue that another round of reviews is needed to properly assess the quality of the new content.
> > > >
> > > > Aside from the remaining issues, here is a list of remarks concerning the newly added content:
> > > >
> > > > i. None of the added Figures are mentioned in the main text, this reduces the impact of the new material.
> > > >
> > > > ii. Line 161, the index $t$ is used instead of the index $i$ to define $\bar{\alpha}_t$.
> > > >
> > > > iii. The parathesis in Equation $3.3$ are useless, they can be removed.
> > > >
> > > > Side note: For future submissions, I encourage the Authors to refer to the points made by the Reviewer in the same order as they were raised. This helps to verify that each point has been tackled in the answers to the reviewers.

---

> > > > > ### Author Response · Authors · 2024-11-24
> > > > >
> > > > > Thanks for checking our responses and we are glad that your concerns are addressed. We have addressed the remaining concerns as well in the updated PDF. We have addressed the remaining points from your initial review as well and are working on releasing the code (as promised in the original response).
> > > > >
> > > > > We respond to your latest query below. We would be grateful if you continue to engage with us in a discussion and are willing to revisit your assessment in the light of these revisions.
> > > > >
> > > > > > The requested ablations, the newly added experiment in Appendix C, and the reformulation of many sentences/paragraphs make the content of the updated submission largely different from the original submission.
> > > > >
> > > > > We believe that the newly added experiment and the ablations are all in support of the main claim that the submission was making: our approach is a simple way to train policies of different parameterizations (classes and types) via offline RL and online fine-tuning, and these experiments are largely in support of this claim. In particular, we already showed two real-robot experiments and have added a third one with OpenVLA; the ablations are merely meant to justify design choices of our approach; and the rephrasings are meant to primarily make claims precise; and have made no revisions in the experiments section in the main paper at all, except for minor rewording and a reference to the OpenVLA Appendix. We do not believe that the main contributions of the paper have changed.  **Could you please  _precisely_ clarify what revisions in particular make the content very different from the main submission or change the main contributions of the paper?**
> > > > >
> > > > > Ultimately, per our understanding, the goal of revisions and back-and-forth discussion in ICLR is for the reviewers and the authors to work together towards improving the quality of a submission. The [ICLR reviewing guide](https://iclr.cc/Conferences/2025/ReviewerGuide) explicitly asks reviewers in Step 9 to: “Update your review, taking into account the new information collected during the discussion phase, and any revisions to the submission.“. If the added information is primarily to support the main claims – which we think is the case – the reviewing guide suggests that reviewers should take into account these revisions to update their decision. Hence, we would be grateful if you would be willing to revisit the revision and your final decision in light of our revisions.
> > > > >
> > > > > Alternatively, if revisions or updates were not a concern, could you please let us know what other changes or clarifications will make you comfortable with accepting our paper? Do you find your concerns addressed?
> > > > >
> > > > >
> > > > > Details from remaining issues:
> > > > >
> > > > > - **3**. We corrected the definitions of $V^\pi$ and $Q^\pi$, now in lines 143 and 144.
> > > > > - **4**. We added an else break statement to fix this issue.
> > > > > - **D**. We believe we have now defined all relevant abbreviations.
> > > > > - **N**. We wrote the full Cal-QL objective in Section 3 for clarity.
> > > > > - **O**. We clarified in the algorithm that we require a behavior cloning loss. The specific loss will depend on the base policy parameterization.
> > > > > - **Q**. We rewrote the last paragraph of Section 4.3 for clarity.
> > > > > - **S**. We changed references to the peer-reviewed versions.

---

### Official Review · Reviewer_QUCJ · 2024-11-03

**Soundness:** 1
**Presentation:** 2
**Contribution:** 2
**Rating:** 3
**Confidence:** 5

**Summary:**

This paper trains policies in actor-critic RL with a supervised learning loss on "optimized" actions, i.e., sample actions from actor and pick the top Q-valued actions. This enables fine-tuning diffusion and autoregressive policies with better sample-efficiency than some prior ways to train the actor.

**Strengths:**

- The proposed approach is simple and intuitive, and shows good experiment results on the domains considered.
- The real robot experiments show meaningful improvement with RL fine-tuning, which is a good empirical contribution.

**Weaknesses:**

## Missing important prior works and baselines; lack of novelty
The insight to train the actor with a supervised learning loss is not a new one, and there is a vast literature in actor-critic RL that this paper ignores.

- [1] Neumann, Samuel, et al. "Greedy Actor-Critic: A New Conditional Cross-Entropy Method for Policy Improvement." The Eleventh International Conference on Learning Representations.
They propose an alternative update for the actor in actor-critic algorithms: "N ∈ N actions are sampled according to a proposal policy, the actions are sorted based on the magnitude of the action-values, and the policy is updated to increase the probability of the maximally valued actions".

- [2] Simmons-Edler, Riley, et al. "Q-learning for continuous actions with cross-entropy guided policies." arXiv preprint arXiv:1903.10605 (2019).
They learn a policy to minimize a squared error to the maximal action found by CEM (approximate global optimization). "CGP is a multi-stage algorithm that learns a Q-function using a heuristic Cross-Entropy Method (CEM) sampling policy to sample actions, while training a deterministic neural network policy in parallel to imitate the CEM policy. This learned policy is then used at inference time for fast and precise evaluation without expensive sample iteration"

- [3] Shao, Lin, et al. "Grac: Self-guided and self-regularized actor-critic." Conference on Robot Learning. PMLR, 2022.
"propose a self-guided policy improvement method by combining policy-gradient with zero-order optimization to search for actions associated with higher Q-values in a broad neighborhood."

- [4] Pourchot, Aloïs, and Olivier Sigaud. "CEM-RL: Combining evolutionary and gradient-based methods for policy search." arXiv preprint arXiv:1810.01222 (2018).

- [5] Kalashnikov, Dmitry, et al. "Scalable deep reinforcement learning for vision-based robotic manipulation." Conference on robot learning. PMLR, 2018.
While QT-Opt does not train an explicit actor network, it is a viable method to pick top Q-valued actions. This along with several other actor optimization methods (like listed above) are important prior work.

Based on these, it is clear that PA-RL idea of "sampling and training on top Q-value actions with supervised learning" is not novel. If there is indeed a significant difference, then this paper should compare against the above works as baselines.

## There is no "global optimization", it is just sampling
Why is the first action optmization step labeled as global optimization, when it is just sampling k actions? If I understand correctly, there is no need to evaluate or rank the Q-values of these samples, since anyway local optimization step is performing gradient updates to these samples, and then the policy is trained proportional to the Q-values of the locally-optimized actions. The algorithm is much simpler than it is shown to be. Being simpler would actually make the algorithm more applicable and transparent.


Additionally, the sampling step seems to do the most work in Table 4, instead of the local optimization step. This makes it more likely that a principled approach of training with CEM-searched actions (like in Greedy AC or CGP) performs even better, because it would be real global optimization.


## Soundness concern: No ablation to justify not imitating the max Q-value action
Section 4.2 labels actions by first sampling them according to the Q-values, but there is no ablation to show that the simpler alternative of taking highest Q-valued action is worse. Why would sampling be better when it adds noise to the policy improvement step? Not only does this lack any thereotical basis, it goes against the idea of generalized policy iteration, where the policy must be updated in the direction of arg max of Q-values.

## No convergence guarantees on the resultant RL algorithm
While I intuitively agree that supervised learning on an optimal action would lead to policy improvement, since this paper modifies the RL algorithm, it must have a theoretically guarantee that this change does not lose the convergence guarantees of the underlying algorithm it modifies in the tabular setting. Another alternative is to build on the guarantees of the prior works I have mentioned above. Though, the novelty of this paper would still remain unclear.

## No experiments on PA-RL + standard actor-critic RL like SAC and TD3
The most common actor-critic RL algorithms are SAC and TD3, but this paper does not show how PA-RL improves (or maintains performance) over these algorithms. I appreciate the results on offline and offline2online RL, but to justify this paper's general claims about "actor-critic RL", it should also be shown on off-policy online RL algorithms.

**Questions:**

Raised questions on novelty, prior work, lack of important baselines, soundness under Weaknesses.

---

> ### Author Response · Authors · 2024-11-19
>
> Thank you for your review and the constructive feedback. To address your concerns we clarify the novelty and motivation behind this work below, and have added several new experiments to understand the utility of global and local optimization and several ablations. We also provide comparisons with CEM-based policy optimization akin to QT-Opt, and self-imitation learning to justify novelty of the work. As mentioned in the global note to all reviewers, we also add a new result showing that PA-RL can improve the success rate of OpenVLA, a 7 billion parameter generalist policy via autonomous fine-tuning in the real-world by **75%** in the real world within 40 minutes of interaction. **Please let us know if these responses and edits to the paper address your concerns, and if so, we would be grateful if you are willing to raise your score.** We are happy to answer any remaining questions and engage in a discussion.
>
> > **Novelty**: The insight to train the actor with a supervised learning loss is not a new one
>
> Thank you for pointing out several related works, we have now added these papers in our related work section and cited and discussed them. That said, we would like to clarify that **we never claim** to be the first one to improve a policy with supervised learning and we have updated the paper to remove any wording that might otherwise suggest so.
>
> Instead, the focus of our work is on improving policies of multiple classes in the offline RL and online RL fine-tuning settings. We believe that prior work has not shown results showing that supervised objectives for policy training can be effective with diffusion and autoregressive policies, in offline RL, online fine-tuning, and hybrid RL, and with real-world fine-tuning of generalist policies.
>
> To address this concern, we have now added several results comparing against the baselines and prior work listed above: (1) we add a result showing that naive self-imitation learning (Oh et al. 2018), which optimizes weighted likelihood on actions from the replay buffer, is worse than PA-RL. While this difference from an algorithmic standpoint seems small, as we show in Figure 14 of the updated paper, this makes a dramatic difference in performance; (2) we add comparisons to CEM-based policy optimization from QT-Opt in Appendix D.4 (as we also discuss below), and find PA-RL to outperform this approach too.
>
> **To summarize,** the novel contribution of our method then, is a formulation that allows using any policy, no matter their size (our new result with OpenVLA shows we can fine-tune 7B robot policies using Cal-QL + PA-RL) or parameterization (diffusion or categorical autoregressive policies), to “initialize” a gradient-based search for actions that maximize the critic. The optimized actions then serve as targets for updating the policy itself. **We believe that no prior work has shown a single method to finetune several policies (including large pre-trained policies), and hence, we believe that it is still of interest to the community**.
>
> > There is a vast literature in actor-critic RL that this paper ignores
>
> Thanks for pointing out these works, we have now cited and discussed these works in the related work section. Besides different motivations, goals, and the scope of the results – i.e. these papers with the exception of QT-Opt focus on fully online RL, study small feed-forward policies, purely in simulation, whereas PA-RL also works well in the offline RL setting, scales to 7 billion parameter policies, and performs well on real robots  – we also note several methodological differences between the works mentioned in the review and PA-RL.
>
> Most of these works run some CEM optimization to optimize the policy. In QT-Opt (which also studies the offline RL and online RL fine-tuning setting), this CEM process is initialized from a random unit Gaussian policy. In contrast, PA-RL uses the base pre-trained policy (which is not Gaussian) as the proposal distribution. We posit that in offline settings, with relatively low coverage offline data, it is also important to obtain action targets for imitation that are close to the support of seen actions in the dataset as opposed to randomly sampling actions whose value is high to prevent exploitation and overestimation of the critic. While imitating the offline dataset to initialize the base policy If the base policy is trained with imitation learning, this effectively constrains the search space to be close to seen actions, as long as we take small enough gradient steps.

---

> > ### Author Response · Authors · 2024-11-19
> >
> > **How does PA-RL compare to CEM (akin to QT-Opt)?** To justify the above hypothesis, we ran new experiments in which the diffusion policy is replaced with a CEM optimizer, and the critic is trained with Cal-QL as in our method. Please refer to **Figure 12** in the paper. We observe that for all tested environments, the performance after pre-training (i.e. at step 0, before taking any online steps) is at or close to 0, and while the performance improves over fine-tuning, it remains well below PA-RL with a diffusion policy.
> >
> > | Task                      |   Ours  | CEM + Cal-QL |
> > |---------------------------|:-------:|:------------:|
> > | kitchen-partial-v0        | 78 → 94 | 5 → 57       |
> > | kitchen-complete-v0       | 59 → 90 | 1 → 62       |
> > | kitchen-mixed-v0          | 67 → 77 | 2 → 55       |
> > | antmaze-medium-diverse-v2 | --      | 10 → 99      |
> > | antmaze-large-diverse-v2  | 73 → 95 | 2 → 0        |
> >
> > **Why does CEM not work well?** We plot the difference between predicted Q-values of actions found by CEM, and the Monte-Carlo discounted returns that those actions actually got, in a task in which the dataset contains optimal actions (kitchen-complete-v0, although the trends are maintained for all kitchen environments.) **Please refer to Figure 13 (left).** (https://imgur.com/2VmOEXZ) We observe that at the beginning of fine-tuning, the predicted Q-values are much higher than the MC returns, even much higher than the predicted Q-values further into training, when performance is actually good. This points to the fact that the **CEM optimizer is able to find actions that maximize the Q-function, but are exploiting the Q-function.** In contrast, such actions that the critic is erroneously high at will never be found when using PA-RL where the pre-trained policy proposes action candidates.
> >
> > **How does CEM compare computationally with PA-RL’s action optimization procedure?** Since the standard CEM algorithm initializes optimization from a unit Gaussian instead of sampling from a pre-trained policy, sampling actions with CEM should be significantly faster. However, if we were to initialize CEM’s optimization with samples from a policy, that could bring it to parity with our method with respect to compute time.
> >
> > ***Overall, these results highlight the importance of sampling the base policy as PA-RL prescribes in favor of standard CEM from a random policy for offline RL and online fine-tuning problems, justifying the efficacy of our approach.***
> >
> >
> > > There is no "global optimization", it is just sampling; there is no need to evaluate or rank the Q-values of these samples
> >
> > We want to clarify that our global optimization procedure is indeed a zero-order optimization procedure because it samples multiple actions from the base policy and retains the ones that attain the highest Q-value. It is correct that since these action samples are combined with local optimization, one could, in principle, not discard a small subset of action samples, but this sort of a procedure allows us to reduce computation time. For example, multiple action samples from the base policy can be cached after every epoch of policy supervised learning, but the gradient steps need to be taken for every action candidate and for each action in the batch. Discarding most of the bottom action candidates can speed up training.
> >
> > We believe that this approach is akin to any heuristic-based search method – for example, beam search discards certain candidate prefixes of a sequence to speed up search, in contrast to retaining all prefixes till the end and then picking the best one. Akin to this, discarding samples speeds up training as well.
> >
> >
> > The reason why we use the term “global optimization” is because of the relation to zeroth-order optimization and because we wanted to contrast it against local optimization (gradient updates) which only updates each action in a local ball around an action sample. **We agree that the terminology can be confusing here, and are open to modifying the terminology to make the contribution precise if the reviewer has suggestions.**

---

> > > ### Author Response · Authors · 2024-11-19
> > >
> > > > **Soundness concern:** No ablation to justify not imitating the max Q-value action.
> > >
> > > We thank the reviewer for requesting further analysis. To address this concern, we have performed new ablation experiments and add the following two tables to the paper:
> > >
> > > 1) PA-RL with Diffusion policies either selecting the distillation targets with argmax over Q-values, or with a softmax categorical distribution over actions.
> > >
> > > | Environment         | Argmax (%)  | Softmax (%) |
> > > |---------------------|:-----------:|:--------:|
> > > | kitchen-partial-v2  | 89      | 95  |
> > > | kitchen-complete-v2 | 90     | 95 |
> > > | kitchen-mixed-v2    | 68    | 75 |
> > > | CALVIN              | 61     | 47  |
> > >
> > > 2) Standard deviation of the Q-values of action candidates ($\widetilde{\mathcal{A}}^T_{\pi, m}$)
> > >
> > >
> > > | Environment         | STD of action candidate Q-values    |
> > > |---------------------|:-----------------:|
> > > | kitchen-partial-v2  | 1.56               |
> > > | kitchen-complete-v2 | 2.66               |
> > > | kitchen-mixed-v2    | 11.54              |
> > > | CALVIN              | 0.02               |
> > >
> > > **Results:** For kitchen environments, selecting action targets from a categorical distribution outperformed the argmax version by 7% on average, whereas on CALVIN the argmax version outperformed the categorical version by 30%. Note that on CALVIN, after the pre-training phase, action candidates have on average a much narrower variance of Q-values than on kitchen domains, leading to a degenerate case of imitating all actions equally if we mimic the softmax distribution over Q-values, without tuning temperature precisely. The results we report on the paper indeed took the argmax over action candidates.
> > >
> > > **Practical workflow:** The above results indicate that a practitioner does not need to commit to one particular approach for their run. Instead, we propose a simple workflow to decide the best performing alternative of using the argmax action vs the soft distribution over all actions by keeping track of the variance of action candidate Q-values during pre-training, and deciding whether to take the argmax if there’s very small variance, or keeping a categorical distribution to maintain better exploration otherwise. We have now added this discussion and the new results in the paper.
> > >
> > > > Theoretical justification
> > >
> > > We already provide a conceptual (mathematical) comparison of PA-RL with a method based only on global action sampling and mimicking the action with the highest Q-value in Lines 313-325 in the paper. This shows that with high probability, we would expect PA-RL to make more aggressive updates than simply sampling $m$ actions, which already is more aggressive than AWR or SIL which often can “overfit” and reduce down to imitation learning when the data is narrow, no matter what the weighting on $\log \pi(a|s)$ is. (see Park et al. NeurIPS 2024). This helps us conceptually explain the importance of local and global optimization in contrast to these prior methods.
> > >
> > > While most theoretical analysis of RL algorithms is performed in a tabular RL setting with discrete actions and states, we do note the notion of gradient steps is ill-defined in such a setting. When only accounting for global optimization only, PA-RL in a tabular setting should resemble a generalized policy improvement operator, and should inherit convergence guarantees of this operator. We will add a formal version of this discussion in the paper.
> > >
> > > > No experiments on PA-RL + standard actor-critic RL
> > >
> > > Please note that we never claimed that PA-RL is a general actor-critic method, we target the paper towards offline RL and online fine-tuning primarily. We do provide results for hybrid RL (i.e., put offline data in the replay buffer of an online RL algorithm) in Section 5.1, where we do find it to help improve performance.
> > >
> > > [1] Byrd, Jonathon, and Zachary Lipton. "What is the effect of importance weighting in deep learning?." International conference on machine learning. PMLR, 2019.
> > >
> > > [2] Xu, Da, Yuting Ye, and Chuanwei Ruan. "Understanding the role of importance weighting for deep learning." International Conference on Learning Representations, 2021.
> > >
> > > [3] Hansen-Estruch, Philippe, et al. "Idql: Implicit q-learning as an actor-critic method with diffusion policies." arXiv preprint arXiv:2304.10573 (2023).

---

> > > > ### Author Response · Authors · 2024-11-21
> > > >
> > > > Dear reviewer QUCJ,
> > > >
> > > > We wanted to check in on whether our responses, updates on the paper, new ablations and baselines addressed your concerns, and whether you had a chance to see the OpenVLA experiment we added. We would be happy to discuss further.
> > > >
> > > > Thank you!

---

> > ### Comment · Reviewer_QUCJ · 2024-11-22
> > **Disagree with certain points in author rebuttal and the claimed contributions**
> >
> > Thank you for your detailed response and new experiments. Several of my concerns are not addressed, especially, the clarification of claims and lack of comparison with important literature.
> >
> > > we would like to clarify that we never claim to be the first one to improve a policy with supervised learning
> >
> > I disagree with this since the abstract explicitly mentioned: "Our insight is that a universal supervised learning loss can replace the policy improvement step in RL, as long as it is applied on “optimized” actions."
> >
> > In accordance with this key insight, there are several prior works that I listed, and a lot of them can be directly applied to multiple policy classes, precisely because they also rely on sampling.
> >
> > > Please note that we never claimed that PA-RL is a general actor-critic method, we target the paper towards offline RL and online fine-tuning primarily.
> >
> > I disagree with this since the title and abstract are written to indicate a general method. In the abstract, there is not a single mention of offline RL or online fine-tuning.
> >
> > I think the paper needs significant revision in claims and writing, if it is indeed centered around offline RL and online fine-tuning. It still remains unclear what exactly is the key technical contribution in that case?
> >
> > > To summarize, the novel contribution of our method then, is a formulation that allows using any policy, no matter their size (our new result with OpenVLA shows we can fine-tune 7B robot policies using Cal-QL + PA-RL) or parameterization (diffusion or categorical autoregressive policies), to “initialize” a gradient-based search for actions that maximize the critic.
> >
> > This is not at all what the paper originally claimed. "Initialization" of gradient-based search with a pre-trained actor is a natural thing to do, so I don't even think this is a novelty. Even for the sake of the argument of novelty, there are significant issues: (a) all results to justify this "initialization" claim are added in rebuttal, so the paper needs to be re-written to justify this as the central novelty and re-reviewed to check if the experiments actually demonstrate this, and (b) I am not even sure if this is a valid novelty because it is the natural way to initialize your search of actions.
> >
> >
> > I would advise the authors to actually read the papers I listed above, as there is a large literature that is ignored, and there are potentially better ways to incorporate supervised learning, which should be thoroughly compared via experiments.
> >
> >
> > ----
> >
> > Quick question about CEM results. Why not do an iterative CEM search with initialization as the actor? Shouldn't that perform better than the current sampling + local gradient optimization procedure, because CEM is a global optimizer? Also, I'd advise the authors to conduct a similar experiment as Figure 13 with offline actor initialization.
> >
> > While I appreciate the empirical result on real robots and OpenVLA in rebuttal, the current paper writing gives a misleading account of its novelty, and the newly claimed novelty about initialization in rebuttal is not justified by the current paper writing or the experiments. Unless I misunderstood something here, I will maintain my rating.

---

> > > ### Author Response · Authors · 2024-11-22
> > >
> > > Thank you for reading our responses and engaging in a discussion with us. Regarding the scope of our claims, we believe that throughout the paper, we limited our claims to “fine-tuning” policies: the title specifically mentions “fine-tuning multiple policy classes”; the first paragraph of the introduction talks about “RL vs imitation learning”, which does not make sense unless we are talking about improving from offline data; the contributions paragraph says (“Our main contribution is PA-RL, a single approach for offline RL and online fine-tuning policies with different parameterizations and classes“); and all of our experiments are in offline RL, online fine-tuning or hybrid RL settings.
> > >
> > > That said, we have done a careful pass through our paper to edit some claims and would love it if you could tell us if the claims look correct in scope and if there’s any particular claim that gives an impression of incorrect scope, perhaps claims regarding the papers you mention. We would appreciate your feedback here in improving the paper – please let us know if these responses address your concerns.
> > >
> > > > I think the paper needs significant revision in claims and writing, if it is indeed centered around offline RL and online fine-tuning. It still remains unclear what exactly is the key technical contribution in that case?
> > >
> > > The technical contribution of the approach is still what we claim – showing that a particular approach of using supervised learning losses can help us train and fine-tune multiple policy classes with offline RL and online fine-tuning.This can be done regardless of scale, parameterization or output type. We show some of the first results with fine-tuning generalist policies, attain state-of-the-art results, and provide a simple approach.
> > >
> > > We believe that this scope of our technical contribution is in line with several prior works including works showing “Offline RL on Diverse Multi-Task Data Both Scales and Generalizes”, which does not build a novel method but shows that scaling offline RL is possible; “Stop Regressing: Training Value Functions via Classification for Scalable Deep RL”, which shows that using an existing loss function in the right way can improve performance and scaling substantially in RL. These works were accepted by ICLR and ICML respectively, with an oral presentation. More generally, we believe that showing an approach that attains state-of-the-art results and is broadly applicable should be of value to the RL community.
> > >
> > > > I disagree with this since the abstract explicitly mentioned: "Our insight is that a universal supervised learning loss can replace the policy improvement step in RL, as long as it is applied on “optimized” actions."
> > >
> > >
> > > We believe that even our previous revision of the paper had changed it to “basic idea” instead of “insight”, which we think does not give an impression of claiming something now. Is there a different choice of wording that you would like here?
> > >
> > > > In accordance with this key insight, there are several prior works that I listed, and a lot of them can be directly applied to multiple policy classes, precisely because they also rely on sampling.
> > >
> > > We never say that these prior methods cannot be applied to multiple policy classes, but are unable to find any empirical results showing so in these prior papers. Please let us know if we missed something. We believe that we have also compared PA-RL to various relevant baseline approaches that are relevant for diffusion policies and autoregressive policies. We think it is unreasonable to expect us to invent extensions of these prior approaches and compare to them since they have not been studied with these policy classes. That said, we have cited and discussed these works in the paper and are happy to change the wording of that discussion as you would like.
> > >
> > > Moreover by this logic, “Diffusion Q-Learning (DQL)” should not have by accepted by ICLR, since it only “applied” SAC with diffusion policies; likewise “SAC” should not have been accepted by ICML since it only applied TD3 with a stochastic policy, but these prior works were still judged to be significant contributions and have had substantial impact. Specifically DQL builds an approach or a system to get RL working with another policy class than Gaussian policies, and that is similar to the nature of our contribution (but smaller in scope than our contribution).

---

> > > > ### Author Response · Authors · 2024-11-22
> > > >
> > > > > I disagree with this since the title and abstract are written to indicate a general method. In the abstract, there is not a single mention of offline RL or online fine-tuning.
> > > >
> > > > The title explicitly says “Fine-tuning” and we have updated the abstract to remove any phrasing that might give a wrong impression about scope. Let us know if there is any other change that you will like.
> > > >
> > > > > This is not at all what the paper originally claimed. "Initialization" of gradient-based search with a pre-trained actor is a natural thing to do, so I don't even think this is a novelty. Even for the sake of the argument of novelty, there are significant issues: (a) all results to justify this "initialization" claim are added in rebuttal, so the paper needs to be re-written to justify this as the central novelty and re-reviewed to check if the experiments actually demonstrate this, and (b) I am not even sure if this is a valid novelty because it is the natural way to initialize your search of actions.
> > > >
> > > > We apologize for the lack of precision in this statement. Rather than “novel contribution”, we mean that our method is able to use any policy class for proposing actions and uses the resulting optimized actions to train that policy. We propose to update this as follows. Please let us know if this works for you.
> > > >
> > > > “To summarize, PA-RL presents a formulation that allows using any policy, no matter their size (our new result with OpenVLA shows we can fine-tune 7B robot policies using Cal-QL + PA-RL) or parameterization (diffusion or categorical autoregressive policies), to “initialize” a gradient-based search for actions that maximize the critic. The optimized actions then serve as targets for updating the policy itself. We believe that no prior work has shown a single method to finetune several policies (including large pre-trained policies), and hence, we believe that it is still of interest to the community.”
> > > >
> > > > > Quick question about CEM results. Why not do an iterative CEM search with initialization as the actor? Shouldn't that perform better than the current sampling + local gradient optimization procedure, because CEM is a global optimizer? Also, I'd advise the authors to conduct a similar experiment as Figure 13 with offline actor initialization.
> > > >
> > > > Initializing the CEM optimization with a fixed pre-trained multi-modal policy would result in averaging different modes of behavior. We hypothesize this will result in CEM considering actions not seen in the dataset, and thus might still suffer from value overestimation. That said, we will run this experiment and get back to you before the rebuttal period ends.

---

> > > > > ### Author Response · Authors · 2024-11-25
> > > > >
> > > > > Dear reviewer QUCJ,
> > > > >
> > > > > We have added the results of the CEM + pre-trained policy initialization experiment to the paper (**Figure 14**). PA-RL resulted in **42%** better offline-only performance across tested domains. During online fine-tuning, CEM + pre-trained policy initialization did catch up and reach the same final performance in 3 out of 5 domains. However, in the other 2 domains (kitchen-complete-v2 and CALVIN), which are more challenging due to low coverage over actions and a multimodal data distribution, PA-RL substantially outperformed CEM on the pre-trained policy, with **66%** and **172%** better final performance respectively. Multimodal data composition and low coverage over actions are properties that we would expect in real-world offline datasets. As mentioned in our previous response, we expect multimodality to hurt CEM due to averaging over different modes of actions, resulting in out-of-distribution actions with narrow “play” data like datasets.
> > > > >
> > > > > **Please let us know if this addresses your concerns, and if so, we would be grateful if you are willing to revisit your assessment.** Thanks so much!

---

> > > > > > ### Comment · Reviewer_QUCJ · 2024-12-02
> > > > > >
> > > > > > Thank you for your response and commitment to adding several experiments. I have read the entire discussion, rebuttal, and the paper again, and I still have 2 primary concerns about the paper.
> > > > > >
> > > > > > 1. The proposed method needs to be validated against methods in prior work. This paper proposes a new way to find "optimized" actions, while there exists a vast literature in that.
> > > > > >
> > > > > > > We never say that these prior methods cannot be applied to multiple policy classes, but are unable to find any empirical results showing so in these prior papers. Please let us know if we missed something. We believe that we have also compared PA-RL to various relevant baseline approaches that are relevant for diffusion policies and autoregressive policies.
> > > > > >
> > > > > > Many of these prior works I cited came before the era of diffusion or autoregressive policies. That does not mean their ideas are not valid to these policy architectures. If tomorrow there is a new policy architecture, then that would not eliminate the contributions of the method proposed in PA-RL, right? One would naturally expect a comparison to find optimized actions with the method proposed in this work. Similarly, these prior methods must be compared to as baselines, because there are already valid ways that are obviously compatible to various policy architectures, something that PA-RL claims as its novelty. It is very much possible PA-RL is a more compatible method to diffusion / autoregressive policies. But, how can one know that without seeing that empirical evidence?
> > > > > >
> > > > > > > We think it is unreasonable to expect us to invent extensions of these prior approaches and compare to them since they have not been studied with these policy classes.
> > > > > >
> > > > > > Therefore, I disagree with this statement and would say it is totally reasonable to expect this.
> > > > > >
> > > > > >
> > > > > > > Moreover by this logic, “Diffusion Q-Learning (DQL)” should not have by accepted by ICLR, since it only “applied” SAC with diffusion policies; likewise “SAC” should not have been accepted by ICML since it only applied TD3 with a stochastic policy, but these prior works were still judged to be significant contributions and have had substantial impact. Specifically DQL builds an approach or a system to get RL working with another policy class than Gaussian policies, and that is similar to the nature of our contribution (but smaller in scope than our contribution).
> > > > > >
> > > > > > While I don't like to engage with the logic of why a previous paper was accepted or not, I never claimed that PA-RL could not be a meaningful contribution. Yes, I agree SAC is more suited to Gaussian policies and is not directly compatible with various policy architectures. PA-RL is solving a meaningful problem, but the way to solve it must be adequately validated.
> > > > > >
> > > > > >
> > > > > > Besides, during the rebuttal, the authors switched their technical contribution to focus on "initialization", which was again switched back to the generality of policy classes in the previous comment. I don't know how to review the paper with that information. Again, that might be a valid contribution, but in that case, the paper should be focused on that claim.
> > > > > >
> > > > > > Anyway, my primary reason for maintaining my score is explained explicitly above. I hope this helps the authors in revising their paper.

---

### Official Review · Reviewer_RHLn · 2024-11-04

**Soundness:** 3
**Presentation:** 3
**Contribution:** 3
**Rating:** 6
**Confidence:** 4

**Summary:**

This paper proposes Parameterization-Agnostic Reinforcement Learning (PA-RL) by addressing challenges in existing RL methods that often tailor their designs to specific policy types, leading to instability or poor generalization when adapting to new ones (e.g., diffusion models or transformers). The main contribution is in optimizing actions directly through a combination of global and local optimization processes, followed by supervised learning updates that train the policy to imitate these optimized actions. This approach simplifies the training, while maintaining stability and sample efficiency across different policy parameterizations. PA-RL achieved good performance in both simulated environments and real-world robotic tasks.

**Strengths:**

1. The paper studies a valid and important challenge, and addresses a major problem in current RL methods. Therefore, I believe the contribution can be impactful.

2. The proposed method is straightforward, intuitive and practical. To my knowledge, the procedure for global and local optimization of actions based on the critic appears to be novel.

3. The experimental results are comprehensive and are in favor of the method. The inclusion of real-world robotics experiments enhances the credibility and practical relevance.

**Weaknesses:**

1. The local optimization step needs further investigation,  as it is the primary distinction of this work compared to previous methods such as self-imitation learning [1]. I recommend that the authors provide plots and detailed results specifically for the gradient-based step. The implementation aspect is significant since computing $\nabla_a Q(s, a)$ requires forward mode automatic differentiation (AD), which is less common in practice and can be computationally expensive depending on the action space and model parameters. Additionally, it is unclear whether this gradient would be effective unless the Q function is well-trained. Even in that case, the gradient may not always indicate the direction toward optimal actions. Visualizing this gradient on toy examples would be beneficial for readers to better grasp its behavior.

2. The method is representing the policy as a categorical distribution over the “good” action candidates $\hat{\mathcal{A}}_{\pi, m}$ which is then optimized with supervised learning. It is unclear to me how this works exactly because the action candidates are changed from iteration to iteration.

Note: I am willing to consider increasing my score if these comments are adequately addressed.

[1] Oh, Junhyuk, et al. "Self-imitation learning." International conference on machine learning. PMLR, 2018.

**Questions:**

1. Could you include results or plots that illustrate the gradient-based optimization of the actions?
2. Did you use forward mode (AD) to compute $\nabla_a Q(s, a)$? If so, were there any specific challenges or intricacies in its implementation?
3. How do you manage the changes in the action set $\hat{\mathcal{A}}_{\pi, m}$ as it evolves between iterations?

---

> ### Author Response · Authors · 2024-11-19
>
> Thank you for the review and constructive feedback on our work. To address your concerns, we have now added new experiments to understand the utility of gradient-based optimization over actions (Appendix D.3) and find that this step is effective especially when the dataset does not exhibit a high-coverage distribution over actions. We also answer the rest of the questions below. **Please let us know if your concerns are addressed, and if so, we would be grateful if you would be willing to raise your score.** We are looking forward to the discussion.
>
> **Clarification on novelty and significance.** We first want to clarify that the focus of our work is on improving policies of multiple parameterization in offline RL and online RL fine-tuning. We believe that prior work has not shown results showing that supervised objectives for policy training can be effective with diffusion and autoregressive policies, in offline RL, online fine-tuning, and hybrid RL, and with real-world fine-tuning of generalist policies.
>
> In regards to self-imitation learning (Oh et al. 2018) specifically, we believe there are some key distinctions between this method and PA-RL and have now added a result showing that **SIL, which optimizes weighted likelihood on actions from the replay buffer, is not enough for obtaining fast improvement in Figure 14.** PA-RL approach samples multiple actions from the base policy instead and is able to optimize the policy far more effectively. While this difference from an algorithmic standpoint seems minor, this makes a substantial difference in performance.
>
> The poor performance of SIL / AWR with diffusion policies is also consistent with prior works ([1], [2], [3]) that show that training expressive models to optimize importance weighted objectives can be problematic, since these models can increase the likelihood of all training points, diminishing the effect of the importance weights. We believe that a simple modification on existing algorithms that leads to substantially improved results is of value to the community.
>
> To summarize, the novel contribution of our method then, is a formulation that allows using any policy, no matter their size or parameterization or output type, to “initialize” a gradient-based search for actions that maximize the critic. The optimized actions then serve as targets for updating the policy itself. **We believe that no prior work has shown a single method to finetune several policies (including large pre-trained policies), and hence, we believe that it is still of interest to the community**.
>
> | Task                     |   Ours  | Self-Imitation Learning + Diffusion Policy |
> |--------------------------|:-------:|:------------------------------------------:|
> | antmaze-large-diverse-v2 | 73 → 95 | 0 → 0                                      |

---

> ### Author Response · Authors · 2024-11-19
>
> ## Questions and weaknesses
> > I recommend that the authors provide plots and detailed results specifically for the gradient-based step.
>
> Thanks for the suggestion. In addition to directly comparing against SIL (as discussed above), we have now added a new experimental result in **Figure 9** of the revised paper to present an ablation for the number of gradient steps for local optimization (https://imgur.com/bm8w3Vk). We analyze two environments that sit on opposite sides of the data coverage spectrum: **(a)** CALVIN, which features relatively little coverage over actions, since the provided dataset is “play data”; and **(b)** antmaze-large-diverse-v2, which consists of high-coverage actions (i.e., coverage as measured by delta x, delta y, which is more relevant to the task).
>
> We find that CALVIN benefits significantly from increased gradient steps in local optimization, getting up to 20% increase in performance after fine-tuning compared to no gradient steps. On Antmaze, performance already reaches close to 100% even without any gradient steps as long as global optimization is used. We hypothesize that because of the high-coverage, using global optimization with a large-enough number of samples from the base policy already recovers good actions (See **Figure 10** for additional ablation for the number of samples from the base policy, which shows sampling at least 5 actions is critical to have strong final performance on antmaze-large-diverse-v2).
>
> Hence, the local optimization is important, especially in real-world robotic datasets that are typically collected via tele-operation as a result of which they exhibit smaller coverage.
>
> > Did you use forward mode AD to compute $\nabla_a Q(s,a)$?
>
> No, we did not use forward mode auto-differentiation here. Our implementation is based on Jax, so we use jax.grad, which uses reverse-mode AD, to take the gradient of the critic values w.r.t. the actions. The full codebase will be provided before the end of the rebuttal period. A similarly straightforward version could be implemented using PyTorch autograd. That said, one could use forward mode AD for computing this gradient as well if needed.
>
> > How do you manage the changes in the action set $\hat{\mathcal{A}}_{\pi,m}$ as it evolves between iterations?
>
> The set of actions $\widetilde{\mathcal{A}}^T_{\pi, m}(s)$ is used as labels for the supervised learning loss for training the policy. These action targets will change after every iteration as the critic also gets updated. Concretely, we do not maintain one fixed set of “good” action candidates – $\widetilde{\mathcal{A}}^T_{\pi, m}(s)$ is regenerated every time we need to do policy distillation, based on the current critic. Please let us know if that does not address your question, and we are happy to clarify more.
>
>
> [1] Byrd, Jonathon, and Zachary Lipton. "What is the effect of importance weighting in deep learning?." International conference on machine learning. PMLR, 2019.
>
> [2] Xu, Da, Yuting Ye, and Chuanwei Ruan. "Understanding the role of importance weighting for deep learning." International Conference on Learning Representations, 2021.
>
> [3] Hansen-Estruch, Philippe, et al. "Idql: Implicit q-learning as an actor-critic method with diffusion policies." arXiv preprint arXiv:2304.10573 (2023).

---

> > ### Author Response · Authors · 2024-11-21
> >
> > Dear reviewer RHLn,
> >
> > We wanted to check in on whether our responses, updates on the paper, new ablations and baselines addressed your concerns, and whether you had a chance to see the OpenVLA experiment we added. We would be happy to discuss further.
> >
> > Thank you!

---

> ### Comment · Area_Chair_NXHh · 2024-11-24
> **From AC.**
>
> Reviewer RHLn: if possible, can you reply to the rebuttal?

---

> ### Comment · Reviewer_RHLn · 2024-11-24
>
> Thank you for your rebuttal and for running additional experiments.
>
> While my main concerns were addressed, I choose to keep my score given the concerns raised by reviewer QUCJ
> .

---

### Author Response · Authors · 2024-11-19

Dear Reviewers,

We sincerely appreciate the time and effort you dedicated to providing thoughtful feedback about our work. Our goal was to build an approach that allows us to fine-tune policies with many parameterizations and scale with offline RL and online fine-tuning algorithms. While the submission already showed results for diffusion and autoregressive policies in offline RL, online fine-tuning, and hybrid RL settings,  we have updated the paper to include several new ablations and results (including fine-tuning of a 7 billion parameter OpenVLA policy using PA-RL + Cal-QL, which to our knowledge is the first result to fine-tune such a large generalist robot policy via value-based online RL fine-tuning in the real world) and have made quite a few edits to address the reviewers’ concerns. We believe that these results have made the paper stronger. We briefly summarize the main changes and this new result of OpenVLA fine-tuning, which is now present in Appendix C of the paper.

## Online fine-tuning of OpenVLA in the real world (Appendix C)

**TL, DR:** PA-RL improves [OpenVLA](https://openvla.github.io/), a 7B-parameter generalist robot policy, by **75%** on a real-world manipulation task after 1 hour of zero-shot language-conditioned trials, and 40 minutes of online RL fine-tuning on the real robot.

**Task and experimental setup:** We consider a new task in the same kitchen environment as our previous two real-world tasks: *“vegetable to sink”*, which requires grasping a toy cabbage and placing it on a plate in the sink. There additionally is a distractor on the scene. We collect 50 rollout episodes by zero-shot prompting OpenVLA with the instruction “put the vegetable on the plate”, and use them to pre-train a Q-function with Cal-QL + PA-RL. Note that while our toy kitchen resembles a kitchen that was present in the training dataset for OpenVLA, the specific task is novel and there are likely significant differences in camera angles and background that affect OpenVLA zero-shot performance. The base OpenVLA model achieves a 40% success rate on this task. The system details of implementation are shown in Appendix C.

**Results:** After pre-training the critic, we run PA-RL + Cal-QL fine-tuning in the real world for 40 minutes (which includes both robot interaction time and OpenVLA training time) with a sparse reward function, manual resets of the environment, and object randomization, similarly to the previous real-robot experiments. The resulting fine-tuned OpenVLA policy obtained a 70% success rate, which is 75% higher than the base OpenVLA, and 40% higher than without the 40 minutes of real-world fine-tuning (i.e., offline only). **We believe that this is the first result that fine-tunes a large generalist policy with 7B parameters with actor-critic RL successfully in the real world.**

## Summary of major changes in the draft

We have added several new experiments as listed below in the updated version of the paper.
- Ablations for the number of gradient steps in Local Optimization (**Figure 9**), and number of samples from the base policy for Global Optimization (**Figure 10**) [Reviewer RHLn, Reviewer Eej1].

- Comparisons to using a CEM optimizer as the policy (**Figure 12**) [Reviewer QUCJ], and to Self-Imitation Learning (**Figure 14**) [Reviewer RHLn].

- Results of PA-RL with Gaussian policies (**Figure 16**) [Reviewer Eej1].

- Plot of performance as a function of wall clock time (**Figure 17**) and discussion about the computational performance characteristics of PA-RL (Appendix E) [Reviewer Eej1].

- Ablation for directly sampling actions from the base policy when calculating Bellman targets, instead of using Action Optimization like PA-RL (**Figure 15**) [Reviewer Eej1].

- OpenVLA fine-tuning results (Appendix C)

---

### Meta-Review · Area_Chair_NXHh · 2024-12-20

**Metareview:**

The authors propose a new way of training a policy in reinforcement learning. The proposed algorithm works by computing a set of "promising" actions using a combination of gradient ascent on the critic and random search. The policy is the learned by doing supervised learning on the "promising" actions. The process is compatible with diffusion.

The main strengths of the paper are: (1) its broad applicability; (2) simplicity of the method (3) good experiments.

However, the paper suffers form serious shortcomings.
- Calling the process used near line 243 "global optimisation" is borderline unsound. A process that samples a small set of actions and evaluates the critic at those actions isn't guaranteed to find a global optimum.
- I am concerned whether a policy optimising equation 4.3 will reach global optimality (this might be a presentation issue).
- The paper has changed too much since the initial submission.
- Reviewer QUCJ made good points about relationship with prior art (especially pre-diffusion).

For these reasons, I will be recommending rejection.

**Additional Comments On Reviewer Discussion:**

There was extensive back-and-forth between the authors and reviewers, which resulted in the authors' adding a substantial amount of experimental results of the paper, to the point that the paper was very different from the original.

In the end, no reviewer was really willing to champion the paper.

---

### Decision · Program_Chairs · 2025-01-22

Reject